# Experimental Investigation of a Helium-Cooled Breeding Blanket First Wall under LOFA Conditions and Pre-Test and Post-Test Numerical Analysis

**Bradut-Eugen Ghidersa** *![ID], **Bruno Gonfiotti** ![ID], **André Kunze** ![ID], **Valentino Di Marcello** †, **Mihaela Ionescu-Bujor**, **Xue Zhou Jin and Robert Stieglitz**

Karlsruhe Institute of Technology, Hermann-von-Helmholtz-Platz 1, 76344 Eggenstein-Leopoldshafen, Germany; Bruno.Gonfiotti@kit.edu (B.G.); mail@ak85.de (A.K.); valentino.dimarcello@lngs.infn.it (V.D.M.); mihaela.ionescu-bujor@kit.edu (M.I.-B.); jin@kit.edu (X.Z.J.); robert.stieglitz@kit.edu (R.S.)
* Correspondence: bradut-eugen.ghidersa@kit.edu
† Current address: INFN, Laboratori Nazionali del Gran Sasso (LNGS), Via G. Acitelli 22, 67100 Assergi, Italy.

**Abstract:** The experimental investigation of a prototypical set-up simulating a loss of flow accident in a helium-cooled breeding blanket first wall mock-up under typical heat load conditions is presented. The experimental campaign reproduces the expected DEMO thermal-hydraulics conditions during normal and off-normal situations and aims at providing some insight into the fast transients associated with the loss of flow in the blanket first wall. The experimental set-up and the definition of the experimental matrix are discussed, including pre-test analysis performed in support of these activities. The major experimental results are discussed, and a procedure of using the acquired data for validating and calibrating the RELAP-3D model of the mock-up is introduced. All these activities contributed to the creation of a relevant theoretical and practical experience that can be used in further studies concerning incidental transients in real-plant scenarios in the framework of DEMO plant fusion safety activities.

**Keywords:** DEMO; first wall; code validation; LOFA; experimental investigation

## 1. Introduction

The Helium Cooled Pebble Bed (HCPB) blanket concept is one of the European Demonstration Power Plant (EU DEMO) blanket concepts currently under development [1]. At the present stage, the blanket concept employs helium as coolant, $Li_4SiO_4$ as breeder material, $Be_{12}Ti$ as neutron multiplier material, and EUROFER as structural material [2–4]. As part of the general strategy [5,6] for the qualification of the DEMO design and its components, in view of licensing and operation, several experiments have been performed or are planned to be carried out in the HELOKA facility at KIT [7,8].

Most of these experiments address issues related to the plasma-facing components, in particular the design and qualification of the First Wall (FW), which faces the plasma and is subject to high heat and neutron loads from the plasma core. Being in front of the breeding zone, the DEMO FW needs to be rather thin to reduce its impact on the reactor breeding ratio while coping with surface heat fluxes ranging from 350 kW/m² in the median region and up to 1 MW/m² in the upper and lower regions.

For HCPB, to manage such loads, one uses high-pressure (8 MPa) helium flowing through small channels imbedded in the FW. As it is inside of the reactor Vacuum Vessel, the integrity of the FW is of paramount importance for the DEMO operation. There are several causes that might hamper the integrity of the FW, but Loss Of Flow Accidents (LOFAs) are among the most dangerous ones because, in the case of a reduced flow, the material temperature increases while the pressure inside the channels remains at levels close to the nominal operating values. Depending on the severity of the flow reduction, the

temperature of the wall facing the plasma can increase to temperatures where the material strength is significantly reduced, placing the FW outside the operating limits foreseen in its design [9].

The studies addressing this kind of accident rely on a combination of different types of numerical tools, the system behavior simulated with MELCOR (Sandia National Laboratories, Albuquerque, NM, USA), or RELAP5-3D (Idaho National Laboratory, Idaho Falls, ID, USA), while local detailed simulations are performed with CFD tools such as CFX or Fluent combined with Ansys Mechanical (Ansys Inc., 275 Technology Drive, Canonsburg, PA, USA) for stress and failure analysis. Despite the significant progress in modeling and the computational capabilities, the numerical simulation of fast transients characterizing a LOFA event still needs experimental data to either benchmark the models or to calibrate certain parameters.

The present paper looks into different details of an experimental campaign carried out in HELOKA to investigate the behavior of a FW mock-up under LOFA conditions. First, the mock-up characteristics is introduced followed by preliminary numerical investigations carried out to define the experimental matrix. The experimental results are discussed afterwards, with emphasis on the calibration and validation the FW model implemented with the RELAP5-3D code as part of the validation of the codes adopted in DEMO safety studies. A best-estimate plus uncertainty methodology are applied to ease the distinction between the user's effects and code limitations (if any).

## 2. First Wall Mock-Up Description

The design solutions of the DEMO breeding blanket are continuously evolving to address various requirements resulting from ongoing studies. For this reason, the manufacturing of mock-ups closely following the design is impractical. For many experimental studies it is then necessary to rely on existing manufactured mock-ups. This is also the case of the present campaign conducted, which employs an existing First Wall Mock-Up (FWMU) manufactured in the frame of the Test Blanket Module (TBM) studies [7].

The FWMU is a EUROFER P92 plate (Figure 1a) in which cooling channels have been manufactured using a spark erosion process developed as part of a study concerning various fabrication paths for a blanket FW [10]. The plate has dimensions of 710 mm × 405 mm × 45 mm. The cooling channels have a square cross-section of $15 \times 15$ mm$^2$ with a 4-millimeter fillet radius at the corners. There are 20 channels in total that are oriented parallel to the long side of the plate. Each two adjacent channels are paired together forming ten U-shaped flow paths (Figure 1c). This solution allows the increase in the length of the flow path from 700 mm to about 1410 mm and, consequently, the increase in the coolant outlet temperature. The wall between two neighboring channels has a width of 5 mm, while the "plasma facing side"—on which the surface heat load is applied—has a thickness of 3 mm. The coolant flows to the channels through circular nozzles (8-millimeter inner diameter). Due to manufacturing reasons, each channel has a horizontal nozzle and a vertical nozzle. The horizontal nozzle is connected at the end of one of the two sides of each channel. The vertical nozzle is attached on the back of the plate at a few centimeters from the end of the channel, thus leaving a dead-end zone on one side of the channel (Figure 1b).

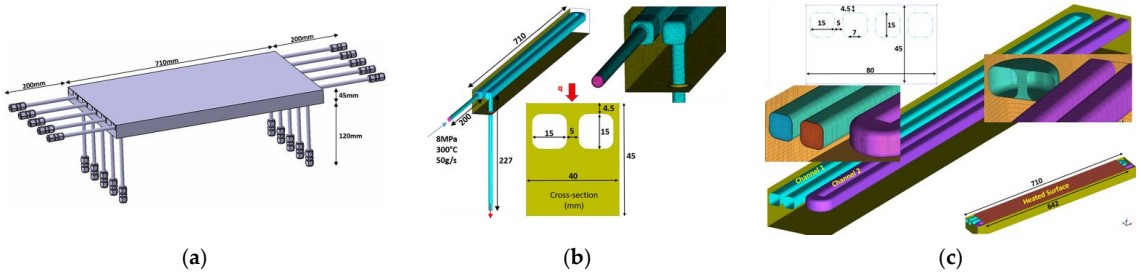

|  (a)  |  (b)  |  (c)  |

**Figure 1.** First wall mockup: (**a**) Mock-up drawing with dimensions; (**b**) Ansys CFX model of a single cooling channel; the arrows at the end of the nozzles indicate the assumed flow direction; (**c**) two channels Ansys CFX model.

The flow characteristic of the mock-up channels was experimentally investigated and reported in [11]. Thus, for designing of the LOFA experiment we had access to the pressure losses in the channels at different flow rates for both flow directions: one using the horizontal (axial) nozzle as an inlet and the other using the vertical one, transverse to the channel flow. The data reported in [11] show a significantly higher pressure drop in channel 1 and 2 as the flow rates are above 30 g/s compared to the values shown by the remaining channels. This is mainly due to pronounced welding seams that reduce the cross-section of the corresponding horizontal nozzles.

## 3. Experimental Set-Up

At the time when the experiment was designed, there were several cooling schemes taken under consideration for the DEMO blanket and FW design [1,3]. One of these schemes considers two independent cooling circuits, i.e., a configuration that should ensure a higher reliability. To simulate such a configuration in HELOKA—which has only one loop—the mock-up channels have been connected to two distinct distribution manifolds (TS-LP-020 and TS-LP-021 in Figure 2). The mock-up channels 1 to 5 are connected to TS-LP-021 while channels 6 to 10 are getting the helium from TS-LP-020. The return flow is also achieved through separate branches (TS–LP-051 for channels 1–5 and TS–LP-050 for channels 6–10, respectively). Each return line is equipped with a flow meter to monitor the flow rates though each branch.

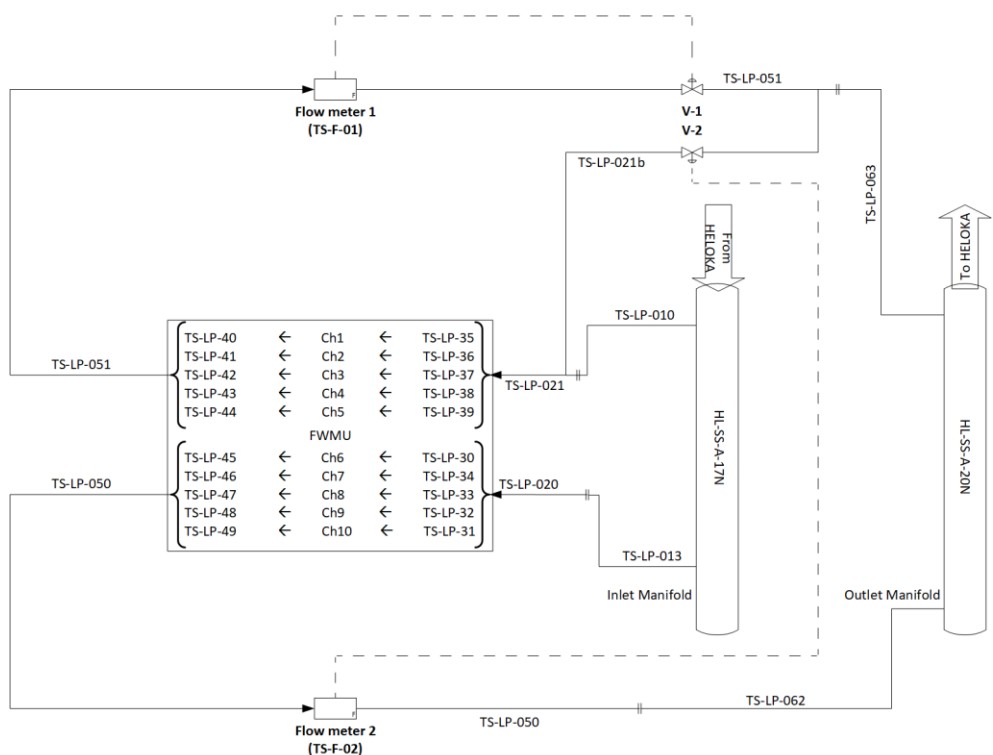

**Figure 2.** Scheme of the preliminary experimental set-up including pipes identification, flowmeters, and valves.

Under this configuration one has the possibility to simulate either a complete or a partial LOFA. The complete LOFA can be achieved either by stopping the circulator or, as was executed in [12], by redirecting the flow through one of the HELOKA bypasses and isolating the test section. In practice, the later method is preferred since the cooling of the mock-up can be achieved faster as compared to the time needed to ramp up the circulator. For simulating a LOFA event on half of the cooling channels (partial LOFA), the opening of the valve V-1 on the return line TS-LP-051 is changed. Because the two flow paths are connected to the same distribution buffers (HL-SS-A-17N for the inflow

and HL-SS-A-20N for the return), closing V-1 would affect the flow rate through the other branch (TS-LP-020 and -050). For this reason, the bypass line TS-LP-021b and the installed valve V-2 are operated to maintain the flow rate through TS-LP-020 at its nominal value independently of the V-1 opening level.

### 3.1. Experimental Matrix and Pre-Test Analysis

At the time when the present study was carried out, the thermal hydraulics of the FWMU had already been simulated for TBM operating conditions, namely 90 g/s per channel with a helium inlet temperature of 300 °C and a pressure level of 8 MPa. While the simulation in [13] assumes that the helium enters and exits the U-shaped channel through horizontal nozzles as well as a uniform heating on the whole mock-up surface, the results indicate that the existence of the 180° bend that leads to a locally increased heat transfer due to the strong mixing associated with the sharp flow turn. The subsequent stress analysis indicates that, given the assumed mock-up fixation, a testing of the mock-up at 0.35 MW/m$^2$ should be possible, while at higher loads, the ratcheting criteria are not met. In these studies, the back side of the mock-up was not allowed to deform in the vertical (Z) direction, and four points, close to the center of the mock-up, were assumed as fixed points. In [7] these studies were continued, examining the possible attachment of the mock-up in more detail. Restricting the length of the heated surface to the area between the insert points of the vertical nozzles and using a specially designed attachment system (attachment F1 in [7]), the study showed that, by leaving the sides of the mock-up free to move, the mock-up would survive even the high heat loads for the assumed lifetime of a TBM set in ITER (850 h operating time and 3000 cycles/year, assuming a total operation of 2.5 years). If the movement on the border of the mock-up is restricted (attachment F2 in [7]), the maximum heat flux, for which the design criteria are met, is reduced.

These previous studies were mainly carried out for TBM-specific conditions and considered higher flow rates and quasi-stationary conditions. To assess if the mock-up can be used to simulate a LOFA event for a DEMO blanket as modeled in [7], dedicated CFD and system codes calculations were carried out. These simulations were also used to determine the operating regime that would provide temperature gradients in the FW cross-section similar to a DEMO breeding blanket. These simulations encompass both detailed thermal-hydrodynamic simulations (only FWMU) using Ansys CFX, as well as system-wise simulations (FWMU and complete HELOKA loop) using the RELAP-3D code.

### 3.1.1. Preliminary Fluid Dynamics and Thermal Simulation of the Test Mock-Up Analysis

The main objective of this analysis was to determine if FWMU can provide relevant information on the behavior of a DEMO blanket first wall under a LOFA event as modeled in [14]. Having U-shaped channels (in contrast to one-way cooling channels for DEMO FW) it is important to select the right inlet and outlet configuration that results in a temperature distribution similar to the that of a DEMO blanket under steady state conditions. Additionally, in terms of dynamic behavior, it is important to check if the mock-up can produce similar transients when the flow rate changes as the one obtained in [14]. In order to clarify these two points, using Ansys CFX, a two-channel model was created to simulate the transient behavior of the temperature field in a blanket FW during a LOFA event: one channel operating at its nominal parameters while the other is subject to a partial (50% flow rate drop from the nominal value) or total LOFA event, similarly to the cases investigated in [14]. To reduce the calculation time for transient simulations, a simplified two-channel model was created (Figure 1c). Thus, by omitting the inlet and outlet nozzles, hexahedral structured mesh could be used for the two-channel model. As it will be discussed later on, this simplification has an impact on the flow and heat transfer of the first segment of the channel (in the flow direction), and the simulation of the second (return) segment was close to the results obtained when the inlet and outlet nozzles were included in model.

The DEMO blanket FW has a single passage counter-flow cooling scheme, the coolant entering the FW on one side and exiting on the opposite side, in each two adjacent channels

helium flowing in opposite directions. For the FWMU, due to the U-shape of the cooling channels, before moving to further investigations, we needed to see which flow configuration provided results closer to the temperature fields calculated in [14]. To clarify this aspect, two different flow configurations have been investigated: a counter-flow pattern where the flows in any two neighboring cooling channels have opposite directions, and a co-current configuration where the coolant in two adjacent flow paths has the same flow direction. In both cases, periodic boundary conditions were applied on both sides of the model to account for the presence of neighbor channels.

The channels were heated by applying a constant heat flux of $300 \text{ kW/m}^2$ on a length of 642 mm on the "plasma-facing" side. This length corresponds to the straight channel length between the 90° inlet and outlet nozzles and the beginning of the 180° bend. For this comparison, the simulation considered steady-state cooling and heating conditions, the same initial conditions assumed at the initiation of the LOFA event. With the coolant flowing in both channels at its nominal values (50 g/s), the temperature distribution on the heated wall in case of counter-flow arrangement was more uniform than the co-flow case, and the surface temperature range was about 10 °C less, as can be seen in Figure 3. In addition to that, the counter-current flow configuration generated a surface temperature distribution closer to the one expected in the DEMO blanket [3,14]; therefore, it was selected as the reference cooling scheme for the experimental campaign. For this reason, only the results for this case are presented and discussed in the following. Note that, as compared to the simulations conducted in [14], for these calculations, the loading conditions are somewhat lower ($300 \text{ kW/m}^2$ as compared to $500 \text{ kW/m}^2$). However, the flow rate was also reduced from 66 g/s to 50 g/s, which resulted in a maximum surface temperature that was only 20 °C lower than in the case modeled in [14].

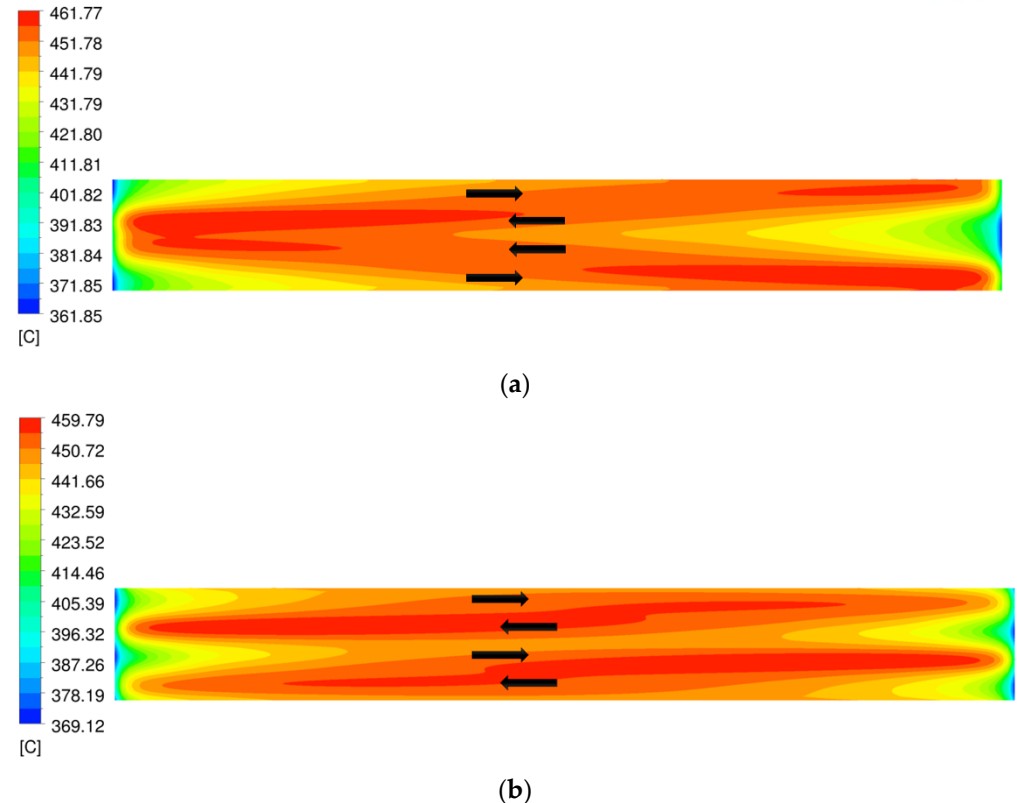

**Figure 3.** Surface temperature distribution for the two channels model at a surface heat flux of $300 \text{ kW/m}^2$: (**a**) co-flow configuration; (**b**) counter-flow configuration. The arrows indicate the coolant flow direction in each channel segment.

Figure 4 shows a comparison of the surface temperature profile for the two-channel model (Figure 3b) with the one obtained for a single-channel simulation that has inlet and outlet nozzles included into the model (Figure 1b). From this figure it can be seen that the absence of the nozzles in the model had an impact mostly on the first leg of the flow path, as can be seen in Figure 4. The profile, taken along a path on the loaded surface that follows the coolant flow, had a more pronounced slope when the nozzle was modeled as compared to the other case. This behavior can be explained by the impact of the entrance effects associated with the change in the cross-section between the nozzle zone and the channel on the heat transfer, as the strong flow mixing in that area locally increased the heat transfer coefficient. However, this is true only for the first leg of the flow path; the profiles on the exit leg (after the 180° turn) showed only minimal differences between the three simulated channels.

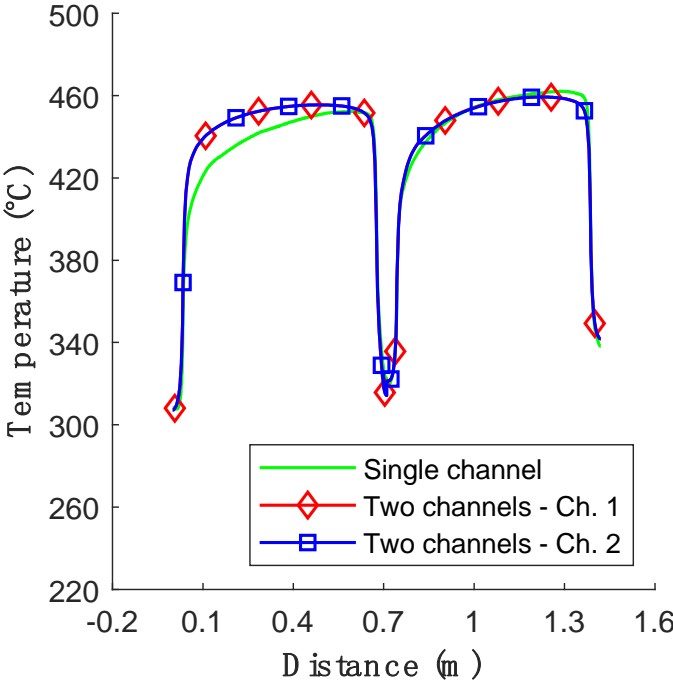

**Figure 4.** Surface temperature distribution for the counter-flow configuration and comparison with the 1-channel model.

The same situation can also be observed in Figure 5, which shows the evolution of the temperature, pressure, and velocity along the channel center line starting from the inlet position of each channel. Thus, looking at the coolant temperature evolution along the channels, it can be seen that, for the two-channel model, the reduced coolant mixing at the inlet, due to the absence in the model of the cross-section change associated with the nozzle-channel junction, moved downstream, at which point the helium in center channel started to heat up. The final helium temperature was, however, only marginally lower for the two-channel model as for the one where the nozzles are included. In both models, passing through the U-turn resulted in an increased flow mixing and, consequently, in an increase in the coolant temperature in the central region. This increase was, however, only local, and the flow recovered its stratified temperature profile about two channel diameters downstream from the bend. This is why the central channel temperature had an apparent drop in value after the sharp increase associated with the 180° turn. When looking at pressure evolution, for the two-channel model, the pressure loss due to the 180° bend was about 6 kPa, which represents almost half of total calculated pressure loss (14 kPa). In comparison with the one-channel model, it can be seen that the pressure in the center of the channel was lower at about 0.1 MPa when the inlet and outlet nozzles are considered, but there was almost no influence on the fluid velocity.

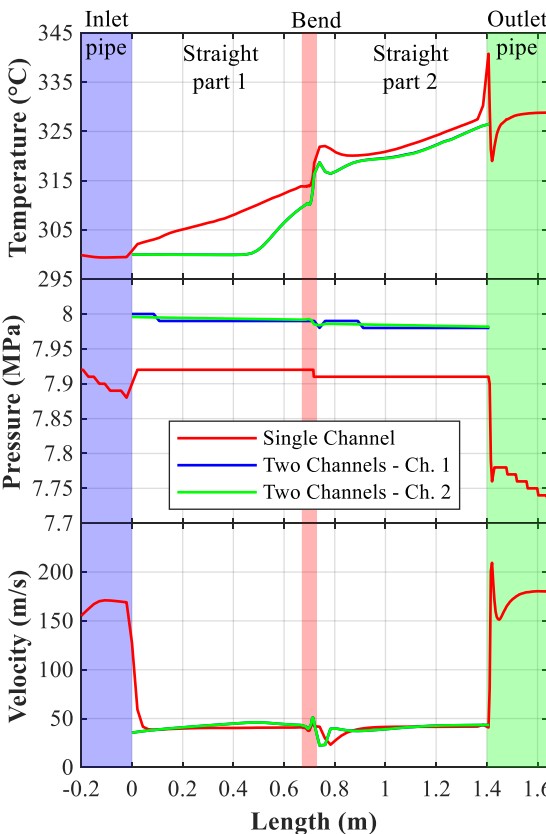

**Figure 5.** Temperature, pressure, and velocity along the channel centerline starting from inlet.

Looking at these two figures one can say that, even though the two models show differences, the trends for the temperature, the pressure, and the velocity are very similar. In particular, for the maximum surface temperature, a parameter of primary interest for the present investigation, the difference was below 1%, the one-channel model temperature being slightly higher (464 °C as compared with 460 °C). As such, one can conclude that the two-channel model introduces only minor deviations as compared with a more realistic, but more computational demanding model that includes the inlet and outlet nozzles. This relevancy of the simpler model is expected to hold also for the simulation of the LOFA transient of interest, the temperature increase being expected to occur over the channel with no or reduced flow, in the area where the transient is dominated mostly by conduction.

Thus, using the two-channel model and the results of the steady-state simulation as initial condition, analyses of two transients were performed: one where the flow rate in the first channel was completely lost in 12 s (total LOFA), and a second case where the flow in the same channel was reduced to half of its nominal value in the same amount of time (partial LOFA). In the second channel the flow rate was maintained at its nominal value during the whole transient.

The imposed time evolution of the flow rate in the LOFA-affected channel is shown in Figure 6a, while Figure 6b shows the resulting temperature increase (surface maximum temperature value).

For the total LOFA, the surface temperature (Figure 6b) reached in the hottest spot at the end of the transient (12 s) is close to 550 °C, which is the current operating limit for the breeding blanket structure. While this does not necessary mean that the FW might fail at this point, it indicates that the risk of failure of the FW increases very rapidly. In case of partial LOFA (mass flow rate reduced to half of the reference value), the rise of the maximum temperature on the top wall surface was less significant, about 27 °C in 12 s, as compared to 90 °C in the case of total LOFA. To determine in a more reliable way the time to failure of the FW in case of a LOFA event, a more exhaustive analysis including mechanical

stress analysis and failure modes analysis needs to be performed, but these topics were outside of the scope of the present investigation. In any case, this analysis suggests that the maximum duration for which the flow is reduced to zero (total LOFA) during the experiment should not exceed 12 s to avoid surface temperatures over 550 °C. When comparing these results with those obtained for the DEMO blanket FW in [1], it can be seen that, for the present case, the temperature rise after 8 s was close to 60 °C as compared to 90 °C in the reference case. This can be explained by the cooling effect on the main flow of the 180° turn area where there is no surface heating. This also explains the apparent "drop" in the temperature profile in Figure 4, where the low temperature area, in the middle of the graph, corresponds to points outside the heated zone. Nevertheless, despite these deviations from the reference case, the simulated dynamic behavior of the mock-up is found to be representative for a first investigation of a DEMO FW LOFA scenario.

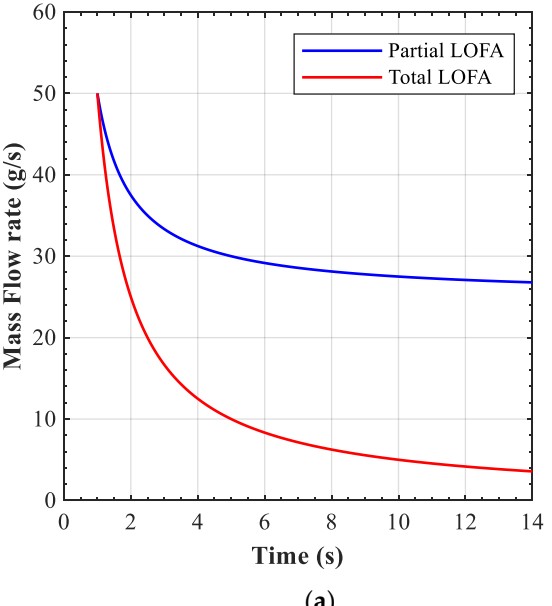

(**a**)

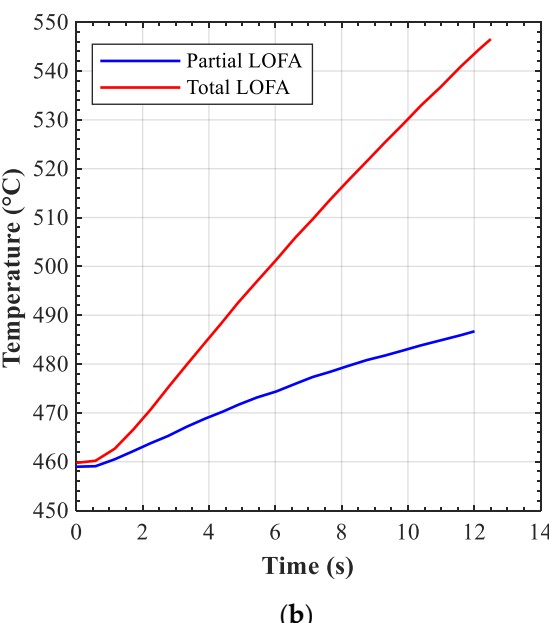

(**b**)

**Figure 6.** LOFA channel flow rate drop (**a**) and maximum surface temperature evolution (**b**) for the partial LOFA and total LOFA.

### 3.1.2. Experimental Set-Up RELAP5-3D Model and Transient (LOFA) Thermal-Hydraulic Analysis

The 3D simulations provided a rather detailed view of the temperature fields within the mock-up, but they tend to become very demanding in terms of computational time for simulating transient behavior, making this approach impractical for investigating a wider range of operating conditions. In particular, the investigation of the dynamics of the mock-up included in the thermal-hydraulic response of the experimental loop was of primary interest when setting-up the experimental campaign. Such a model can provide the information needed for thermal-hydraulics parameters, hints for the finalization of the piping design and components, and it can also ease the definition of the test matrix.

To address these issues, a RELAP5-3D model simulating the mock-up together with the experimental loop was created. For the mock-up model itself, a preliminary calibration of the pressure loss coefficients was performed. In addition, a comparison between the two Ansys CFX models and the RELAP model was performed for the steady-state case with heat load. A detailed description of the RELAP model itself and the calibration procedure can be found in [15]. The nodalization of the large pipes in the test section introduced in Figure 2 is shown in Figure 7. In this model, the pipes connecting the manifolds (TS-LP-021, -020, -050, and -051) and the mock-up had simplified geometries since their detailed design, mostly driven by mechanical stress considerations, was not available at that time. The

valves V-1 and V-2 were represented as control valves (V945 and V944 in Figure 7) with linear characteristics and a flow coefficient ($K_v$) of 25. Flow meters 1 and 2 were modeled as fully opened valves (V946 and V947 in Figure 7) with a concentrated pressure drop coefficient calibrated based on the calculation sheet from the flowmeter manufacturer, and with a flow coefficient of 32.

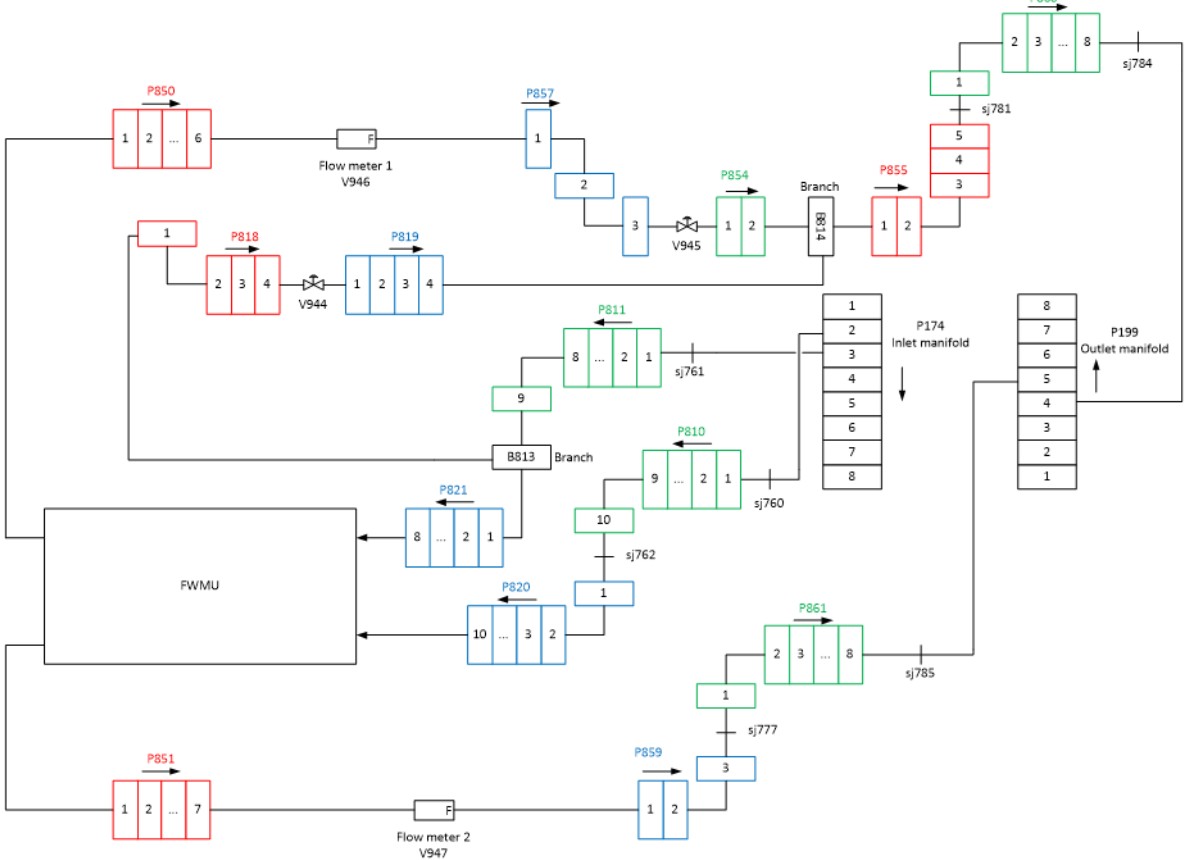

**Figure 7.** RELAP5-3D model of the experimental set-up introduced in Figure 2.

Due to high pressure losses in channels 1 and 2 [11], we decided to carry out the experimental investigations by only heating up the central region of the mock-up surface, i.e., channels 3, 4, 7, and 8 (Figure 8). In the transversal direction (perpendicular on the channels axis), the loading profile in the RELAP5-3D model was constant at 300 kW/m$^2$ along the central channel length, and null on the peripheral channels. The heating profile was also shaped to avoid the heating of the channel's bend region. The LOFA scenario was then simulated starting from a steady-state condition with flow rates in the channels of interest close to the nominal value of 50 g/s (see test section parameters in Table 1). The channels under LOFA conditions are highlighted in yellow in Figure 8.

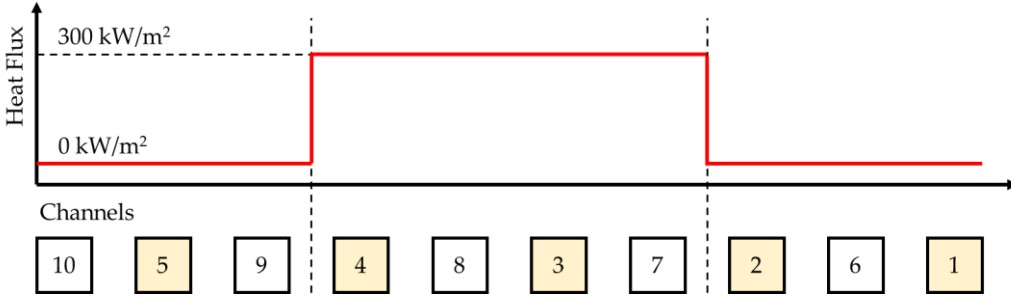

**Figure 8.** Mock-up surface loading.

**Table 1.** Mock-up testing parameters.

| Parameter | Value |
|---|---|
| Total mass flow through the test section (P174) | 466.6 g/s |
| Test section inlet temperature | 305.5 °C |
| Test section inlet pressure | 8 MPa |
| Mass flow through channels 1 to 5—Flow meter 1 (V946) | 218.7 g/s |
| Mass flow through channels 6 to 10—Flow meter 2 (V947) | 247.9 g/s |

From these values it can be seen that having the valve V-1 in one of the flow paths resulted in a lower flow level through the corresponding channels (channels 1–5) even if the flow characteristics of the channels were similar (Table 2).

**Table 2.** Pressure-drop across FWMU (heated) channels.

| Ch. No. | 3 | 4 | 7 | 8 |
|---|---|---|---|---|
| $\Delta p$ (bar) | 1.57 | 1.55 | 1.67 | 1.62 |
| $\dot{m}$ (g/s) | 43.8 | 45.3 | 48.4 | 50.4 |

The fluid (Figure 9) and the heat structure temperature (Figure 10) profiles during the initial steady state were determined by the heat flux shape imposed as a boundary condition. The heated channels (3, 4, 7, and 8) and the related heat structures were subjected to the highest temperature, with channels 3 and 4 having the highest temperature due to the lower mass flow rate crossing them. The sudden drop in the temperature of the heat structure segments corresponding to the channel bend region (Figure 10) was due to the absence of surface heating in that area.

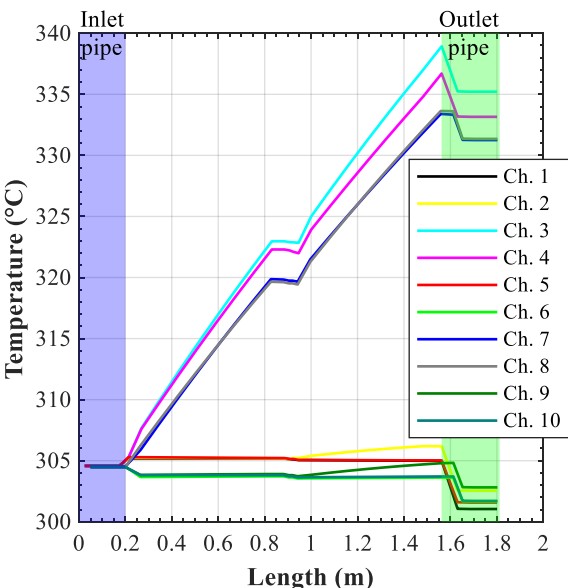

**Figure 9.** Helium temperature along each channel.

The loss of flow was simulated by fully closing V945 valve, which therefore reduced the mass flow rate of channels 1–5. The time for the complete closure of the valve was set to 10 s to simulate the coast-down of the HELOKA circulator. As valve V945 closed, the mass flow rate in channels 1 to 5, which is measured by flowmeter 1, fell to 0 (Figure 11). As a consequence, the flow rate in the other channels (6 to 10, reported by flowmeter 2) tended to increase, but the controller immediately reacted by opening up the bypass valve V944, thus maintaining the flow through flowmeter 2 at the specified level. This transient

also perturbs the inlet manifold mass flow rate, so valve V901 slightly closes to adjust the desired inlet flow. The transient does not significantly affect the pressure drops.

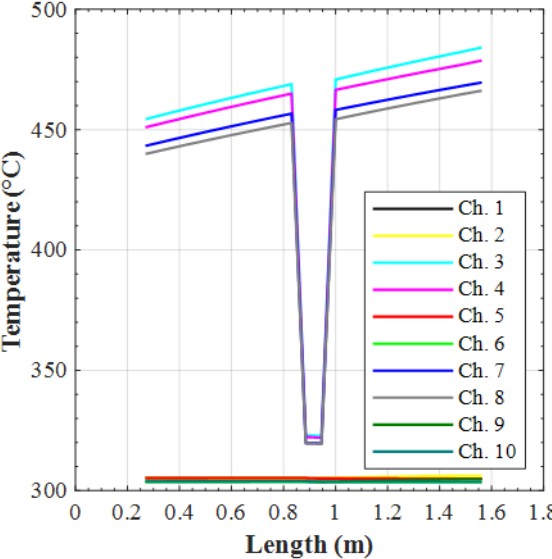

**Figure 10.** Surface mock-up plate temperature for each channel.

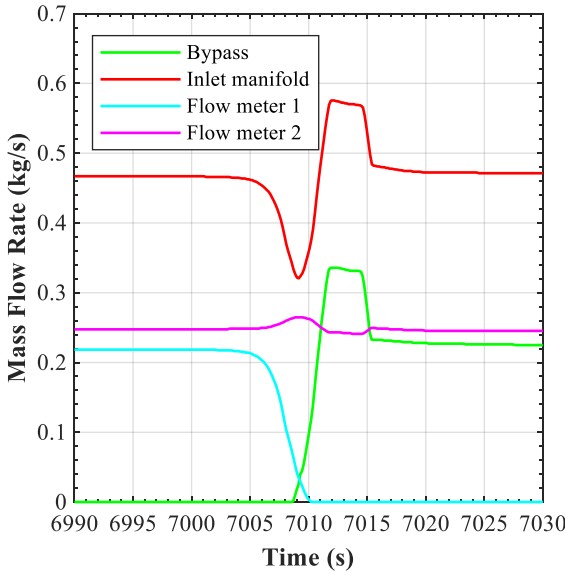

**Figure 11.** Mass flow rate through the inlet manifold, the bypass, and the flowmeters.

As far as the FWMU cooling capabilities are concerned, the LOFA might have caused a significant temperature increase (Figure 12) in channels 3 and 4. This quick increase might be overestimated by RELAP because the heat conduction between neighboring structures is not considered, as discussed in [15]. Since the heat is redistributed across the different channels, a more realistic situation would see lower temperatures for channels 3 and 4, but slight temperature increases in the other channels. In any case, the results suggest that the maximum temperature should not exceed 550 °C in the first 30 s of the transient. In this regard, the figure obtained from the Ansys CFX analysis suggests a much quicker increase (in about 12 s the temperature reached 550 °C). Because of the discordant results, the Ansys CFX figure was taken as a reference for the experiments being the most limiting one.

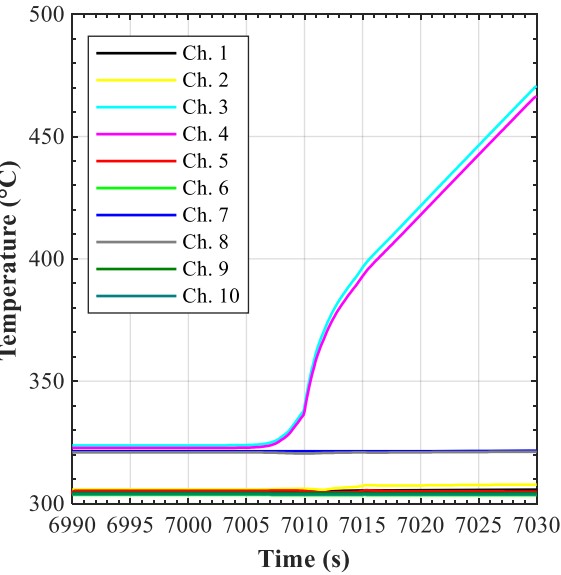

**Figure 12.** Average helium temperature in the FWMU channels.

## 3.2. Final Experimental Set-Up

The results of the pre-test numerical analysis formed the basis for the final set-up of the HELOKA loop. The validity of the cooling scheme was confirmed by the RELAP5-3D analysis, and the final layout of the pipes connecting the manifolds with the FWMU was decided following a dedicated stress analysis (Figure 13). The complex arrangement of these pipes is due to the fact that the pipes had to fit within the lower half of the HELOKA vacuum chamber (3 m in diameter) while maintaining the stress levels below the allowable limits defined by DIN-EN 13480. With the electron beam gun mounted on the top side of the vacuum tank, the upper half of the vessel was kept free of any helium piping to avoid the risk of accidentally heating them up while applying the heat load on the mock-up surface.

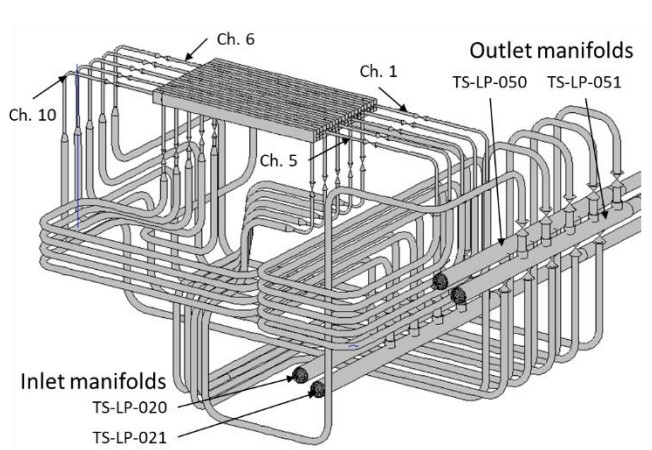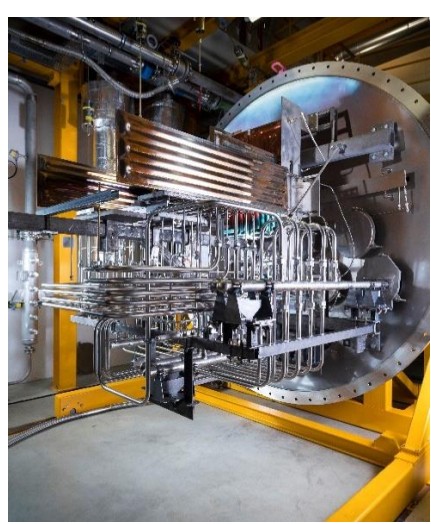

**Figure 13.** Inlet and outlet pipes layout.

To keep the pressure losses in the connecting pipes at a reasonable level, the inlet and outlet pipes vary in diameter along their path from and to the manifolds. Thus, the pipe segments connected to the manifolds have a 21-millimeter inner diameter (1-inch precision pipes). Similar to the mock-up, the size of the pipes had to be reduced to 9 mm inner diameter (1/2 in pipes) to keep within the allowable limits the stress levels in the mock-up nozzles.

At the connection between the 1 in and 1/2 in pipes several sensors were installed as follows:

- Six coolant temperature sensors (TS-T-002, TS-T-003, TS-T-004, TS-T-007, TS-T-008, TS-T-009) were installed at the outlet pipes of channels 2, 3, 4, 7, 8, and 9. The sensors are 3 mm in diameter type K, tolerance class 1. To increase the response time, the sensors were directly immersed in the helium stream.
- The pressure drop (TS-P-006, TS-P-005, TS-P-004, TS-P-001, TS-P-002, TS-P-003) and the outlet pressure (TS-P-016, TS-P-015, TS-P-014, TS-P-011, TS-P-012, TS-P-013) of the six channels of interest (2, 3, 4, 7, 8, and 9) were also monitored. The absolute pressure sensors are of type DMP 320 from BD Sensors GmbH (<0.1% accuracy at a measuring range of 10 MPa) while the differential pressure sensors are Siemens Sitrans P DSIII with a measuring range of 0 to 5 bar (accuracy 0.065% $\times$ 0.5 MPa).
- Two sensors (TS-T-013 and TS-T-018) were clamped on the outside of the channels 3 and 8 inlet pipes, measuring the coolant inlet temperature (1-millimeterthick, type K, tolerance class 1 sensors).
- No dedicated measurements were made for channels 1, 5, 6, and 10.

A complete list of the sensors used for the present analysis is presented in Table 3. The table also includes the accuracy of the sensors as given by the manufacturer. The only exception is the mass flow rate, which was calculated according to DIN-EN-ISO 5167-1 and 2:2004-01 standards from the measurements of the pressure loss over the orifice (TS-P-F01 and TS-P-F02) and the helium density at the pressure and temperature measured by the absolute pressure (TS-P-F11, TS-P-F12) and temperature sensors (TS-T-F01, TS-T-F02), respectively.

**Table 3.** List of helium parameters monitored for the FWMU.

| Parameter | Sensor/Signal Label | Sensor Type | Range Accuracy |
|---|---|---|---|
| Flow rate | TS-F-01 TS-F-02 | Orifice(d = 38.394 mm, β = 0.69474) | 0 to 500 g/s 1.2% $\times$ flow rate (g/s) |
| Absolute pressure before the flowmeters | TS-P-F11 TS-P-F12 | Membrane pressure sensor (Siemens Sitrans P DSIII) | 0 to 10 MPa 0.16% $\times$ 10 MPa |
| Differential pressure over the flowmeters | TS-P-F01 TS-P-F02 | Membrane differential pressure sensor (Siemens Sitrans P DSIII) | 0 to 0.5 MPa 0.065% $\times$ 0.5 MPa |
| Temperature before flowmeter | TS-T-F01 TS-T-F02 | Thermocouple (type K, class 1) | 300 to 1000 °C 0.004 $\times$ temperature (in °C) |
| Inlet and outlet temperature on channels | TS-T-013 TS-T-018 TS-T-002 TS-T-003 TS-T-004 TS-T-007 TS-T-008 TS-T-009 | Thermocouple (type K, class 1) | 300 to 1000 °C 0.004 $\times$ temperature (in °C) |
| Differential pressure over the channels | TS-P-001 TS-P-002 TS-P-003 TS-P-004 TS-P-005 TS-P-006 | Membrane differential pressure sensor (Siemens Sitrans P DSIII) | 0 to 0.5 MPa 0.065% $\times$ 0.5 MPa |
| Absolute pressure at the outlet of the channels | TS-P-011 TS-P-012 TS-P-013 TS-P-014 TS-P-015 TS-P-016 | Membrane pressure sensor DMP 320 from BD Sensors GmbH | 0 to 10 MPa 0.16% $\times$ 10 MPa |

The heat load on the mock-up surface was applied using an electron beam gun. To have a sharp transition at the edges of the heated zone and limit the power deposition

on an area of 580 × 160 mm on top of channels 3, 4, 7, and 8, four water-cooled plates create a mask on top of the mock-up. In addition, an infrared camera (IR camera) recorded the temperature of the heated surface. The IR camera was installed outside the vacuum chamber, and the FWMU image was transmitted toward the camera lens using two mirrors.

### 4. The Experimental Campaign

The purpose of the experimental campaign was to provide a robust set of data and benchmarks for the validation of numerical tools. This was achieved by exploring different thermal-hydraulic configurations and different flow control strategies. The information gathered from the pre-test analysis was used as a basis for the set-up of the experimental campaign.

During normal operation, the total flow through the mock-up was set to 500 g/s or 650 g/s by setting the speed of the HELOKA circulator accordingly, so that each circuit would receive half from the total flow rate. In reality, due to the presence of the valve V-1 and the higher pressure-loss coefficients characterizing channel 1 and 2, the flow does not distribute evenly between the two branches, the flowrate for the channels 1 to 5 representing only 46% of the total flow when the main valve (V-1) was fully (100%) opened.

The LOFA event was simulated by closing V-1 to a predefined value, i.e., similarly to what was simulated with the RELAP-3D model (Section 3.1.2). However, to have a quick response at the initiation of LOFA event, the bypass valve (V-2) had to be kept slightly opened (5%). The reason for this was that, due the valve characteristic, below this value no measurable change in the flow distribution can be observed and the flow through the valve can be assumed as zero. On the other hand, operating V-2 from a 5% opening gives an immediate response in the flow distribution, avoiding the dead time that would occur if the valve would be fully closed. A PID controller was initially tested to control the bypass valve, but it led to oscillations within the HELOKA loop when trying to control the fast opening and closing of the bypass and the main valves. The best results were obtained by setting an optimal V-2 opening for each V-1 closing position. For both valves, V-1 and V-2, the limit for the opening and closing speed was set to 10% per second. When the LOFA event was initiated, the two valves changed their opening to the specified value. These optimum values were determined through several trials, while the mock-up surface was not heated.

The heating of the mock-up surface was achieved by using the electron beam gun. This was set to generate a homogeneous heat load of 300 kW/m$^2$ on an area of 580 × 160 mm of the mock-up surface (see Figure 14). The area corresponds to a surface that covers only the channels 3, 4, 7, and 8. The heat flux was estimated calorimetrically by calculating the helium power intake using the inlet and outlet mock-up temperatures, the flow rate, and the coolant pressure in the circuit. Adding up all the associated uncertainties, the overall uncertainty of the heat flux measurement can be estimated to be about ±15%.

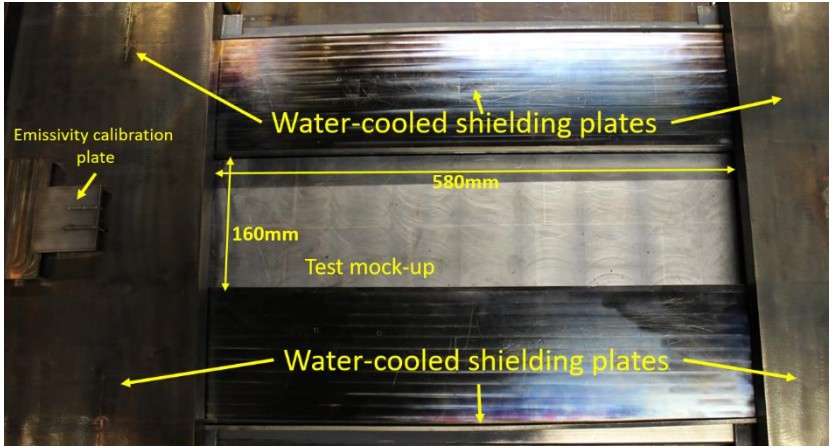

**Figure 14.** View of the mock-up as installed in HELOKA.

The IR camera measuring the surface temperature excursion during the LOFA was set up by imposing two parameters: the emissivity and transmission coefficient. The transmission was always set as 0.33 based on previous calibration experiments. The emissivity was set at the beginning of each experimental session and rechecked at the end of the daily campaign using the procedure described in [1].

The fact that the camera was installed outside the vacuum chamber with the image of the FWMU reaching the camera lens after the reflection onto two mirrors has two main drawbacks:

- The image is inclined and so the thermal shield affects the recordings (Figure 14). The shield hides channels 4 and 8 (the last one only partially). Reflections due to the shield are also shown on top of channel 7 (darker area in Figure 15a), and at the right and left ends of the heated zone (brighter spots in Figure 15a). This is the reason why the temperatures averaged over the rectangular area marked Ch 7_2 in Figure 15a appear to be higher, the actual temperature being in the same range as the one recorded for Ch 8_14.

- The mirrors and the associated structures were left uncooled, and they started to move as their temperature increased. This condition affects the estimation of the emissivity and the transmission factor that change progressively throughout the day.

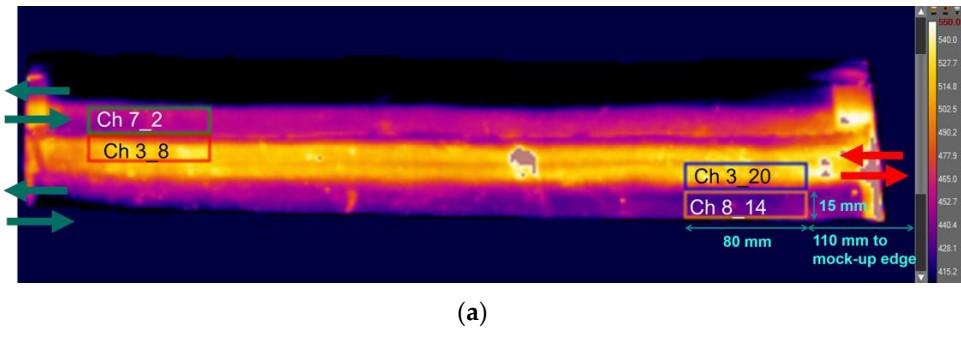

(**a**)

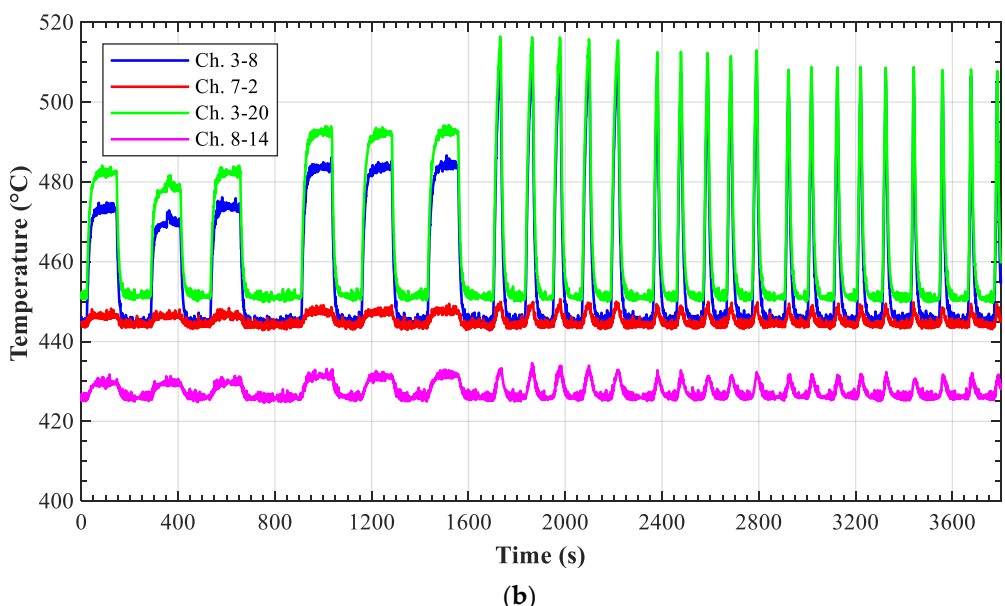

(**b**)

**Figure 15.** Mock-up surface temperature measurement with an IR camera during a run with a 300 kW/m² surface loading: (**a**) snapshot during a total LOFA run (valve V-1 at 0%); the arrows indicate the coolant flow direction in the observed channel segments; (**b**) evolution of the averaged surface temperature during an experimental run.

In addition, the recordings were disturbed by the local non-uniformities of the heated surface emissivity, such as the spot indicating temperatures above the defined scale (bright gray color) on top of channel 3. These spots became more and more evident as the experiment progressed because of the cooling and heating cycles. Adding up all the associated uncertainties the mean uncertainty of the surface temperature should be around $\pm 15\ ^\circ$C, making this measurement more qualitative then quantitative.

Two batches of tests were performed: the first batch at 300 kW/m$^2$ with a total mass flow rate of 500 g/s (Table 4), and a second batch at 330 kW/m$^2$ and 650 g/s (Table 5). Because of the similarities found between the two batches, only the results of the tests at 300 kW/m$^2$ are discussed in the following section. Figure 16 shows the evolution of the flow rates in the two cooling branches during one experimental run, the same one as the one for which the surface temperature shown in Figure 15b is recorded.

**Table 4.** First batch of tests performed at 300 kW/m$^2$.

| Parameter | Steady State | Partial LOFA Runs | | | | | Full LOFA |
|---|---|---|---|---|---|---|---|
| Valve V-1 opening (%) | 100 | 60 | 40 | 20 | 10 | 5 | 0 |
| Valve V-2 opening (%) | 0 | 34 | 44 | 54 | 63 | 64 | 67 |
| LOFA duration (s) | - | 120 | 120 | 23 | 12 | 6 | 3 |
| Flow meter 1 (g/s) | 230 | 140 | 120 | 73 | 53 | 40 | 30 |
| Flow meter 2 (g/s) | 270 | 270 | 270 | 270 | 270 | 270 | 270 |
| Max. outlet temp. (°C) | 325 | 337 | 340 | 348 | 346 | 343 | 334 |
| Avg. pressure drop—Circuit 1 (mbar) | 1730 | 650 | 490 | 195 | 105 | 80 | 50 |
| Avg. pressure drop—Circuit 2 (mbar) | 2140 | 2140 | 2140 | 2140 | 2140 | 2140 | 2140 |

**Table 5.** Second batch of tests performed at 330 kW/m$^2$.

| Parameter | Steady State | Partial LOFA Runs | | | | | Full LOFA |
|---|---|---|---|---|---|---|---|
| Valve V-1 opening (%) | 100 | 60 | 40 | 20 | 10 | 5 | 0 |
| Valve V-2 opening (%) | 0 | 34 | 44 | 54 | 63 | 64 | 67 |
| LOFA duration (s) | - | 120 | 120 | 23 | 12 | 6 | 3 |
| Flow meter 1 (g/s) | 300 | 170 | 150 | 90 | 64 | 56 | 43 |
| Flow meter 2 (g/s) | 350 | 350 | 350 | 350 | 350 | 350 | 350 |
| Max. outlet temp. (°C) | 323 | 334 | 337 | 346 | 344 | 342 | 338 |
| MAvg. pressure drop—Circuit 1 (mbar) | 2935 | 970 | 760 | 260 | 150 | 110 | 65 |
| Avg. pressure drop—Circuit 2 (mbar) | 3550 | 3550 | 3550 | 3550 | 3550 | 3550 | 3550 |

From Figure 15b as well as from Tables 4 and 5 it can be seen that, for 60% valve opening scenarios, the flow rate in the affected branch dropped to 61% from the steady-state condition and to 52% for a 40% valve opening, correspondingly. While the surface temperature of the mock-up increased, the remaining cooling capabilities of the channel 3 and 4 were sufficient to stabilize the surface temperature below 550 °C (Figure 15). The same behaviour can also be observed for the coolant temperature evolution presented in Figure 17. These plots were obtained by averaging over the several runs performed with the same parameters. The associated random uncertainties vary from 0.4 °C to 1.3 °C for the 60% opening case and from 0.3 °C to 1.1 °C, showing good reproductibility of the experiment. The total uncertainty (random and systematic), which is shown in Figure 17, varies from 2 °C to 2.9 °C, the maximum being obtained for the channel 3 outlet temperature for both cases.

The situation was different once the valve V-1 opening was set at lower values as 40%. When closing the valve to 20%, 10%, or 5% the surface temperature approached 550 °C and showed no sign of settling at a value below this limit. For the 20% opening, the trend plotted in Figure 18a indicates that the coolant temperature at the outlet of channel 3 and 4

shows a tendency to settle oppositely to the other two cases (10% in Figure 18b) and 5% in Figure 18c) where the evolution was almost linear for the recorded time.

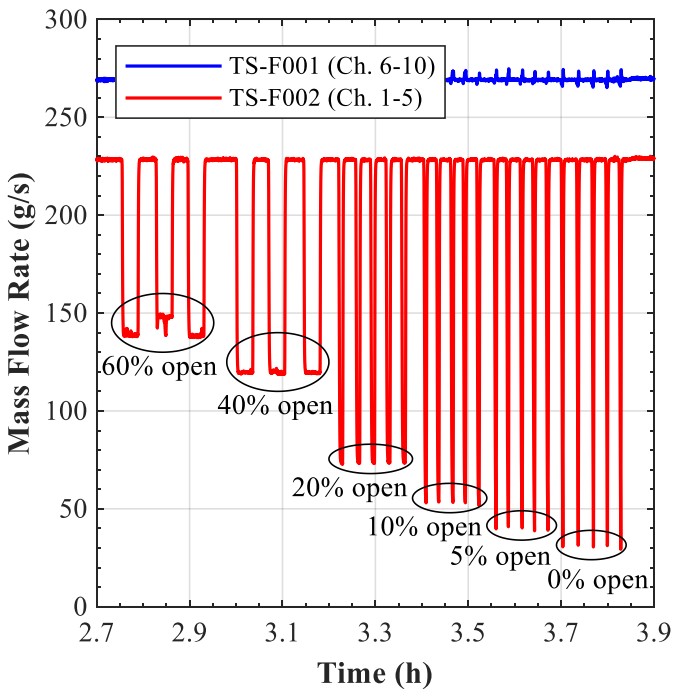

**Figure 16.** Experimental tests execution example (300 kW/m$^2$ heat load).

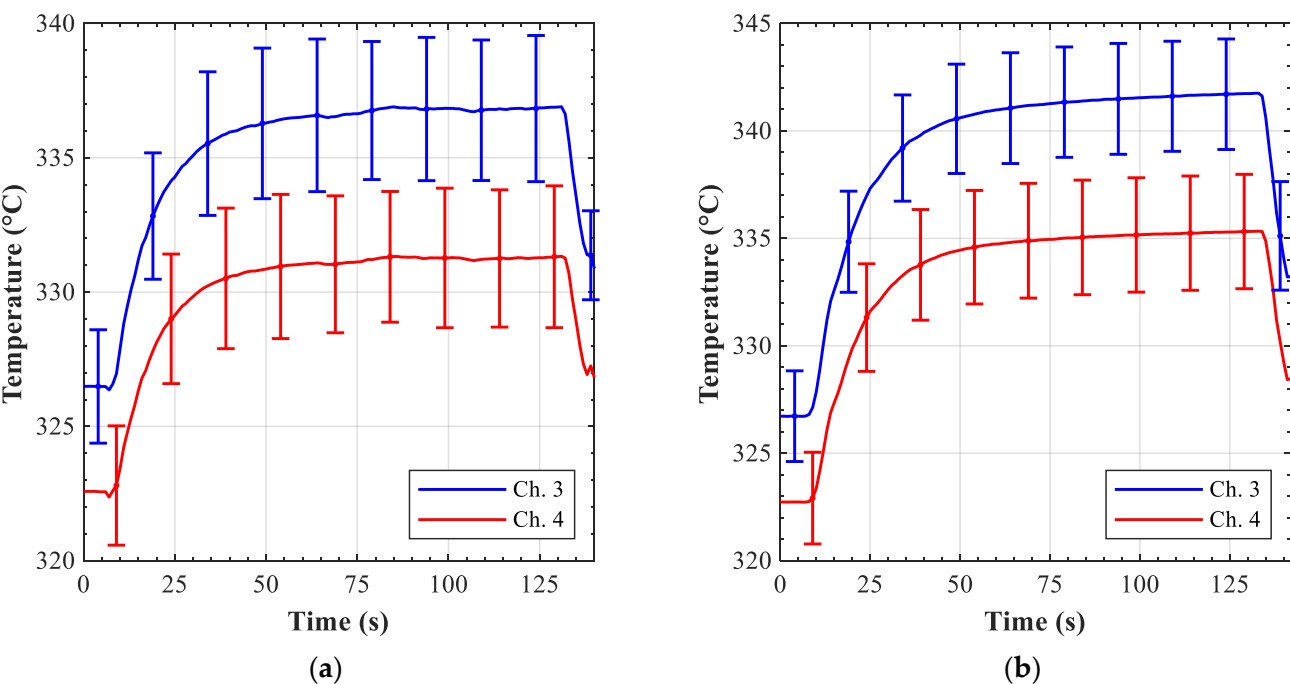

**Figure 17.** Outlet coolant temperature from channels 3 and 4 at a heat flux of 300 kW/m$^2$ averaged over all similar runs at: (**a**) 60% valve opening; (**b**) 40% valve opening. The indicated uncertainties are total uncertainties, which are varying between 2 and 2.9 °C.

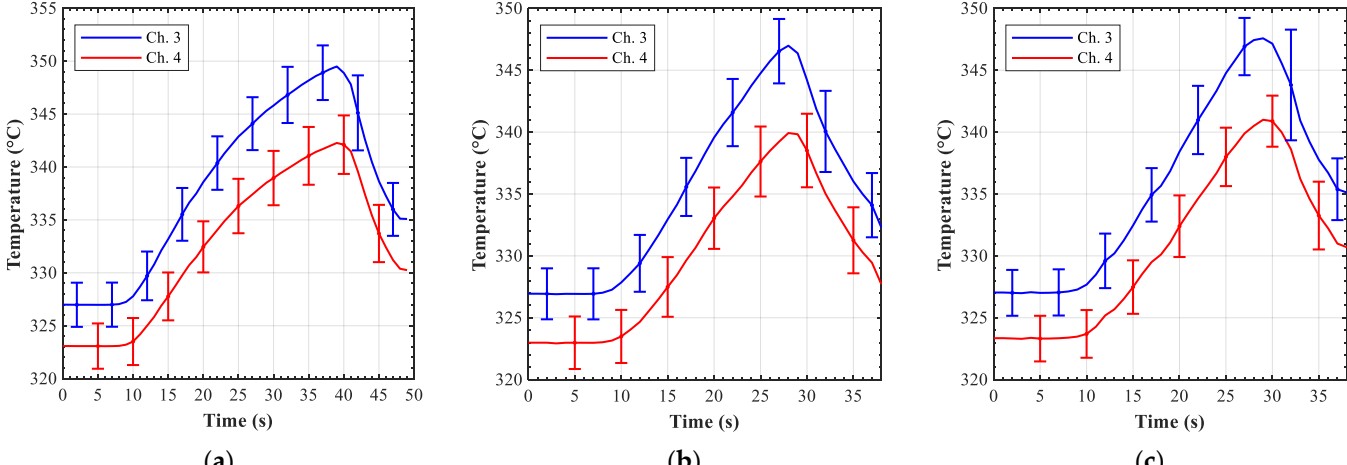

**Figure 18.** Outlet coolant temperature from channels 3 and 4 at a heat flux of 300 kW/m$^2$ averaged over all similar runs at: (**a**) 20% valve opening (uncertainty between 2 °C and 3.5 °C); (**b**) 10% valve opening (uncertainty between 2 °C and 3.4 °C); (**c**) 5% valve opening (uncertainty between 1.8 °C and 4.5 °C).

From the three pictures in Figure 18 it can be seen that the time for which the valve V-1 was maintained at reduced opening was decreased from 30 s in the case of a 20% opening to only 20 s for the 10% and 5% cases. The reason for this is that the simulated LOFA event was terminated (V-1 reopens) once the IR camera measured temperatures over 550 °C on the mock-up surface. Since the transient duration was shorter, the temperature field was not fully established and, while locally, the temperature reaches 550 °C rather quickly, there are regions that have a lower temperature increase rate. This non-uniform heating rate is the explanation why, for the temperatures plotted in Figure 15b, the recorded temperature values in the region Ch 3_20 showed a tendency towards decreasing in their maxima with respect to the valve V-1 opening.

The "total LOFA" case, namely the one where the valve V-1 set point was set to 0% showed some distinct features as compared with the others. First of all, it should be mentioned that, even if the set-point was zero, the flow rate through the channels 1 to 5 dropped only to about 30 g/s (6 g/s per channel in average), as can be seen in Figure 16. The reason for this is that, at the lower end (<1%) of the valve opening, the valve actuator cannot follow the imposed closing ramp of 10%/s. It actually takes a bit longer than 1 s to fully close the last few % down to 0. Since the temperature on the mock-up was rising very quickly, it was not possible to keep the valve at 0% long enough for the flow through the sensor to drop to 0. In addition to that, the flow meter systematic uncertainty at these small flow rates is high, so the actual flow rate could have been lower.

The typical coolant temperature evolution at the outlet of channels 3 and 4 in a 0% scenario (300 kW/m$^2$) is shown in Figure 19. As can be seen the outlet temperature increased in both channels, but between 16 s and 20 s it remained in quasi-steady conditions. Then, it increased again, peaking at 342 °C in channel 3 and 337 °C in channel 4. After 23 s the temperature decreased again because a sufficient mass flow rate is re-established through the channels.

The quasi-steady temperature between 16 and 20 s is probably due to small backflows occurring in these channels. Actually, the coolant in the outlet manifold is colder than that in the outlet of channels 3 and 4, being the result of the mixing of the coolant also coming from the unheated channels (channels 1, 2, and 5). This colder coolant mixes with the residual coolant crossing the channels, showing this peculiar plateau. For the experiment we used a sampling rate of 1 Hz, but, even if we had increased it, the temperature sensors might not be sufficiently accurate to follow the exact evolution of the system. According to the manufacturer data, the temperature sensors should have a response time of 1–1.5 s if sudden variations occur. Considering the experimental conditions and the different coolant (helium instead of air), a response time of about 1.5–2 s may be more realistic. This time is

almost the half of the duration of the plateau (4 s or even less), so the actual temperature of the coolant crossing the channels 3 and 4 might be lower than measured.

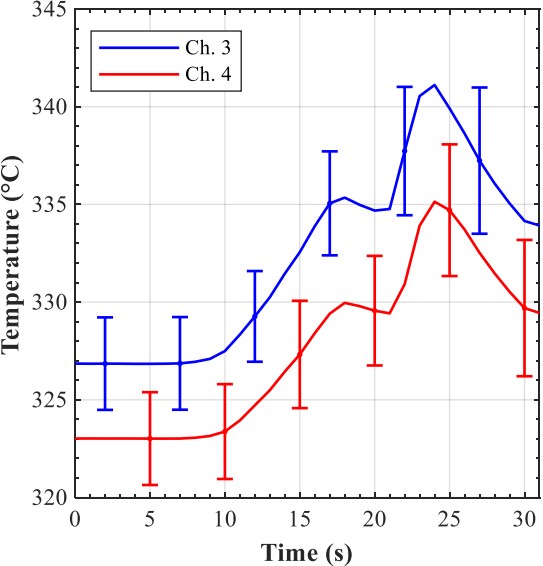

**Figure 19.** Outlet coolant temperature from channels 3 and 4 for 0% valve opening case at a surface load of 300 kW/m$^2$. Uncertainty varies from 2.3 °C to 3.9 °C, the largest value corresponding to the temperature measured at the outlet of channel 3.

However, this plateau is a characteristic of only the 0% valve opening scenario. For the 5%, 10%, and 20% scenarios the total mass flow rate through channels 1–5 decreased but no flow inversion occurred (Figure 18). As a result, the temperature increased up to about 350 °C in channel 3 and to 343 °C in channel 4. Compared to the 0% scenario, the maximum coolant temperature at the outlet in these latest scenarios was higher, but the increase rate was lower due to the longer duration of the transients and to the higher wall–fluid heat transfer.

Even though the Ansys CFX and RELAP5-3D numerical analyses were not executed for comparison purposes, it should be highlighted the good overall agreement found between their results and the experimental ones. In particular:

- The maximum duration of 12 s suggested by Ansys CFX to avoid the overheating of the FW was largely confirmed by the experimental tests.
- The rapid escalation of the helium outlet temperature in the case of the full LOFA scenario suggested by RELAP5-3D (albeit overestimated due to assumptions made) was also confirmed by the experiments.

## 5. Post-Test Numerical Analysis

The brief analysis of the experimental results described in Section 4 confirmed the validity of the main observations obtained with the numerical models used in the pre-test analysis. For the following, a more detailed and systematic approach was used to validate and calibrate our numerical models with the help of the experimental data. As part of the general effort to improve the simulation of the DEMO blanket cooling under abnormal and accidental conditions at a system level, hereafter, we apply a predictive modeling methodology [16] to obtain optimally predicted model parameters and responses, including their (reduced) uncertainties. The adopted best-estimate methodology is a rigorous procedure assimilating all available experimental data and computational uncertainty-afflicted results to provide best-estimate calibrated model parameters and responses together with their uncertainties. This best-estimate methodology eases the distinction between the code's limitations and user's effects while providing valuable information about the influence of each model parameter on the considered model responses [16].

Thus, the data obtained through the experimental campaign as used to validate and calibrate the DEMO breeding blanket first wall model performed using RELAP5-3D code. The preliminary results of this activity are summarized in the reference [17].

The experimental campaign provided a consistent amount of data, but instead of simulating few runs for each valve opening scenario, we decided to simulate "average" tests to reproduce the "average" behavior of the FWMU and not its specific behavior during a specific run.

### 5.1. The RELAP5-3D Model

As compared to the work presented in [15], given the different aim of these post-test and validation analyses, the model includes only the mock-up and the connected pipes extending upstream to the position where the inlet temperature sensors downstream to the flow meters instead of the whole HELOKA loop (Figure 20). Since the helium flow boundary conditions were taken from the experiment, this choice allowed a substantial reduction in the computational time while still preserving the relevant information concerning the experiment. The modeled ancillary pipes are:

- The inlet pipes from the position of the inlet coolant temperature sensors up to the FWMU nozzle (volumes in light green in Figure 20). These inlet pipes are not reproduced for channels 6–10, as the sensor placed on a straight pipe has a length of less than 10 cm;
- The outlet pipes (volumes in gray in Figure 20);
- The outlet headers up to the position of the flow meters (volumes in dark gray in Figure 20).

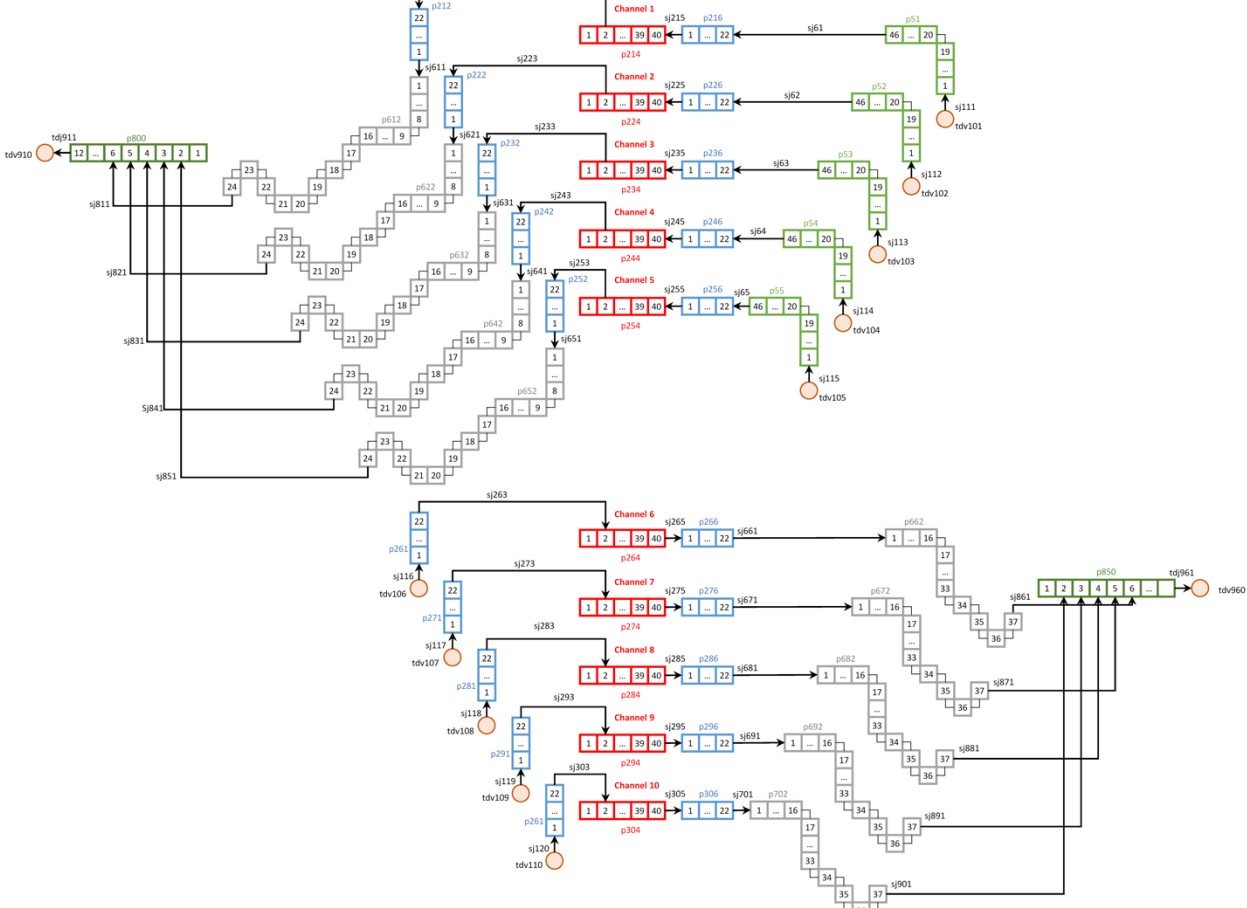

**Figure 20.** RELAP5-3D model adopted for the post-test analysis.

The adopted model mainly consists of pipe components connected together by means of single junctions. The number of nodes of each pipe was chosen after a sensitivity analysis aimed at reducing the influence of the spatial discretization on the results. The inlet conditions of the ten FWMU channels (red and blue volumes in Figure 20) were imposed with ten time-dependent volume, while the total mass flow rate through each branch of the FWMU was imposed by means of two time-dependent junctions reproducing the flow meters on the outlet headers. Because of the few sensors installed at the inlet of the FWMU, the same inlet coolant temperature was adopted for channels 1–5 and 6–10 based on the reading of the sensors installed on channels 3 and 8. The inlet pressure in channels 1 and 5 was assumed as a mean value of the readings from the sensors installed in channel 2, 3, and 4. A similar approach was also adopted for channels 6 and 10, but the values from the sensors installed on channels 7, 8, and 9 were used instead.

Three physical models of the RELAP5-3D code were required to reproduce the experimental tests [17]:

- The hydro-dynamic model to reproduce the pressure drop through each channel. In the 0% valve opening scenario backflows in some FWMU channels occurred because of the low mass flow rate. The magnitude of these backflows depend on the equilibrium between the pressures drops in forward and reverse directions in a single channel as well as the equilibrium between the pressure drops among the five channels belonging to the same inlet and outlet branch.

- The heat transfer between the coolant and the heated surface in steady and transient conditions. Several correlations were available to reproduce this phenomenon, but the Gnielinski correlation [18] was deemed the most appropriate one, as mentioned in [19].

- The conduction of the heat through the thickness of the heated surface (axial heat transfer). As a matter of fact, the heat coming from the electron beam gun was partially removed by the coolant and partially transferred to the non-heated channels because of the conduction occurring through the thickness of the top plate. This phenomenon plays a major role in the LOFA scenarios and can be considered in RELAP5-3D through the activation of the conduction enclosure model.

### 5.2. Set-Up of the Best-Estimate Methodology

The application of the best-estimate methodology passes through four steps. The first step is the selection of the parameters and the responses of interest. For the current scenarios about 100 parameters are considered and categorized as:

- Concentrated pressure drops at geometrical discontinuities;
- Roughness of the pipes walls;
- Thickness of the FWMU face under heat load;
- Material's properties (thermal conductivity and specific heat capacity);
- Inlet pressure of channels 1 and 5 (both affected by the LOFA);
- Input parameters of the conduction enclosure model.

In turn, the responses of interest are the outlet pressure and temperature from channels 2, 3, 4, 7, 8, and 9. However, the temperature sensor on channel 7 showed unreliable values, so it was not actively considered in the analysis.

The second step is the definition of the uncertainty for each parameter. This aspect is one of the most complex ones, because most of the considered parameters are calculated through correlations. The evaluation of the uncertainty of such correlations is not a straight-forward process, and it requires a good knowledge of their applicability range and the conditions for which they were developed for. In this regard, the definition of the uncertainty might be a source of the user's effects, but the numerical framework of the methodology is capable of softening this aspect through the inclusion of the available experimental data.

The adopted uncertainties are summarized in Table 6. These values were obtained from the data from the tests for determining the hydraulic characteristics of the chan-

nels performed on the FWMU [11], and from the data corresponding to the steady-state operation at 300 and 330 kW/m$^2$ acquired during the testing campaign.

**Table 6.** Calculated and assumed uncertainties for the considered parameters.

| Parameter | Uncertainty (%) |
|---|---|
| Concentrated pressure drops at geometrical discontinuities: | |
| • From vertical nozzle to square channel; | ~10% |
| • Square channel 180° turn; | ~5% |
| • From horizontal nozzle to square channel; | ~10% |
| • Other discontinuities. | ~10% |
| Roughness of the pipes walls | ~15% |
| Thickness of the FWMU face under heat load | ~3% |
| Material's properties | ~15% |
| Inlet pressure of channels 1 and 5 | see Table 3 |
| Input parameters of the conduction enclosure model | ~3% |

The third step is the set-up of a strategy to evaluate the influence of each parameter on each response (sensitivity). Several approaches can be followed, but without an analytical solution of the problem only the brute force approach can be applied. This approach is based on the execution of several code runs each characterized by the variation of a single parameter. Usually, each parameter is varied twice; in the first run it is increased by 3–5% and in the second one it is decreased by the same amount. The sensitivity for the specific parameter is then calculated as:

$$S_{r\alpha} = \frac{\alpha^+ - \alpha^-}{r^+ - r^-} \tag{1}$$

where $\alpha$ and $r$ are the parameter and the computed response of interest, respectively. The subscripts + and – denote values assumed (parameter) or calculated (response) in the two runs. The main drawback of this brute-force method is the computational time and the management of the code's runs (twice the number of the parameters selected).

The fourth and last step is the application of the numerical framework described in [16]. The framework will provide a best-estimate value for each parameter and computed response, together with their best-estimate standard deviations. In due regard, if properly applied the methodology might provide an insight of the capabilities of the employed numerical code. If a remarkable difference still exists between the experimental and best-estimate response the cause could be an undue user's effect or an effective code's limitation. Then, through a detailed analysis of the code models and the user's assumptions it should be possible to identify the source of the discrepancy.

*5.3. Results*

As for the pre-test analysis, the FWMU model (only the nozzles and the 180° bend channels) developed for this validation study was initially calibrated against the pressure drop characterization tests discussed in [11]. Then, the full model (Figure 20) was calibrated against the steady-state conditions at 300 kW/m$^2$ and 330 kW/m$^2$.

The model obtained through these preliminary calibration (called "Default run" in the following) was adopted as a starting point for the analysis of the experimental campaign. The six valve opening scenarios characterizing the experimental campaign can be categorized as:

- Scenarios in which the surface under heat load reaches 550 °C and backflows are established in some channels (0% LOFA scenarios);
- Scenarios in which the surface under heat load reaches 550 °C but no backflows are established (5%, 10%, and 20% LOFA scenarios);

- Scenarios in which the surface under heat load does not reach 550 °C (40% and 60% LOFA scenarios).

Among the three categories, the first one is the most challenging to reproduce with the RELAP5-3D model.

The 0% LOFA scenario at 300 kW/m$^2$ is the sole representative of the first category. The first aspect of interest is the evolution of the pressure in the branch affected by the LOFA (channels 3 and 4 in Figure 21a,b correspondingly). In both channels the outlet pressure increased as the velocity of the coolant decreased (LOFA started at 5 s and the valve as fully open again at 26 s), and then returned to the steady-state values once the full mass flow rate is re-established. Four curves were reproduced for both channels:

- The "Exp" curve refers to the experimental results.
- The "Default run" curve refers to the results achieved with the preliminary calibrated input deck.
- The "BE meth." curve refers to the results achieved applying the best-estimate methodology. These results do not come from a code run, but from the numerical framework of the methodology.
- The "BE run" refers to the results achieved with an input deck built using the best-estimate parameter values calculated through the methodology. In contrast to the "BE meth." curve, these results are obtained from a calculation.

The results of the default and BE runs show no substantial differences, suggesting that the default values were already capable of reproducing the transient. The mathematical framework (BE meth. curve) was also in substantial agreement with these runs, and it showed a peculiar reduction in the standard deviations as the mass flow rate decreased because with no—or small—fluid motion the dynamic contribution to the total pressure was low and the static pressure dominated. The good agreement found between the experimental and the computed results for the outlet pressure drop provided the first impression of the validity of the hydro-dynamic model.

Figure 21c,d show the outlet coolant temperature from channels 3 and 4. Compared to the pressure, these results present a more scattered behavior:

- In channel 3 both the BE and the default runs showed a marked drop in concomitance of the plateau, but the BE run was more capable of representing the maximum temperature reached at 23 s.
- In channel 4 no backflows were established in either run, resulting in a poor reproduction of the experimental trend.

Several aspects of the experimental set-up and the assumptions made influenced these curves. The first aspect is the reliability of the measurements. As stated in the experimental campaign section, in this phase the actual coolant temperature might have been lower due to the response time of the sensors while the code reported "instant" responses. The second aspect is the assumption of the inlet pressure of channels 1 and 5. For both channels an average value from the reading of sensors 2, 3, and 4 was adopted to set up the input deck. The third aspect is related to total pressure drop in the forward and reverse fluid direction. When the mass flow rate through a channel approaches zero the fluid direction is strongly influenced by the pressure drops in both direction; if the pressure drop in reverse direction is lower than that in the forward direction the fluid will likely move in the reverse direction. The pressure drops in both fluid directions were calibrated at the very beginning of the validation process (the pressure drop characterization tests [11]), but those in "reverse" direction were not further calibrated in the subsequent pre-calibrations (steady-state conditions of the LOFA campaign) because of the lack of experimental data.

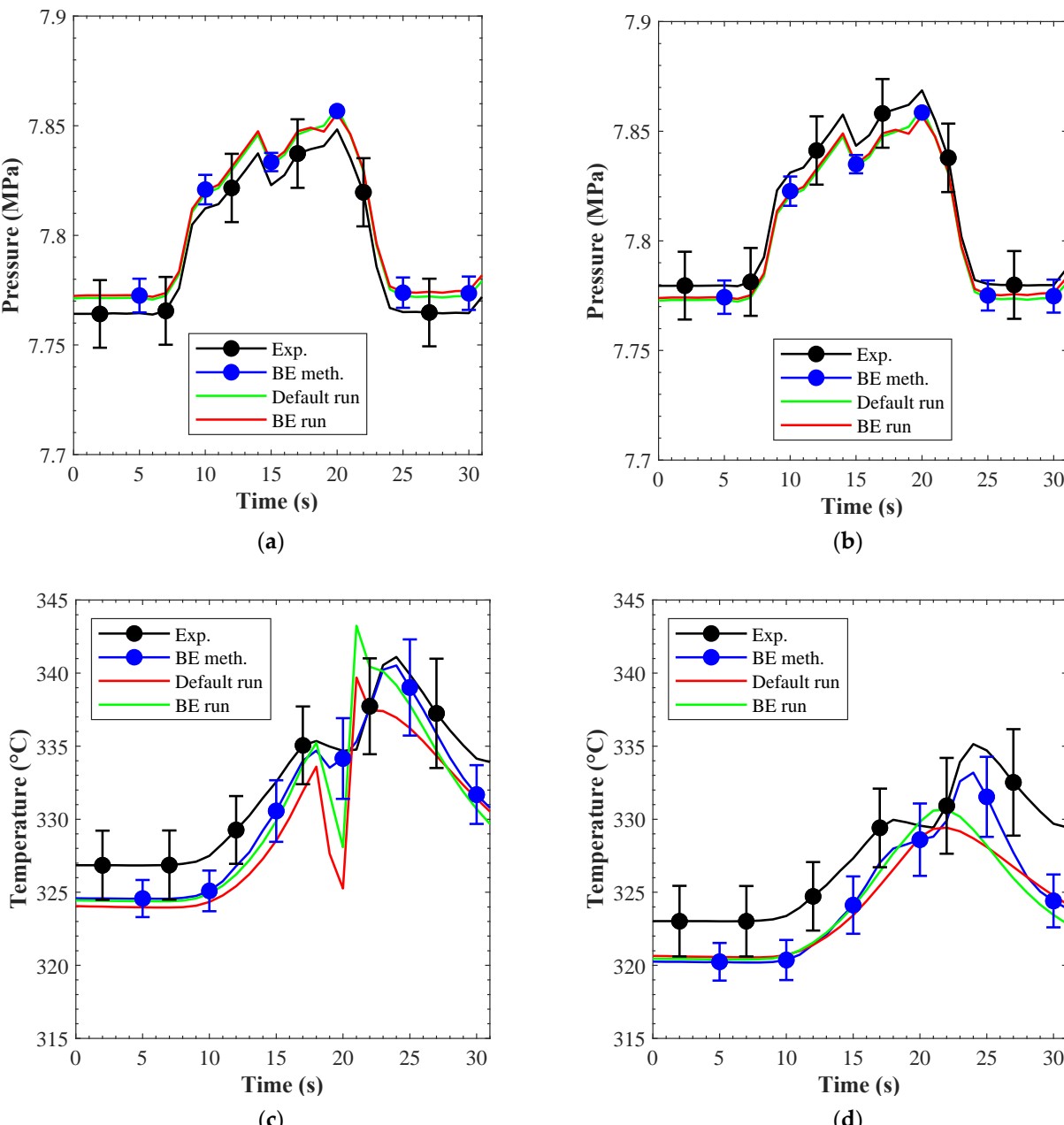

**Figure 21.** Helium parameters evolution during a 0% valve opening scenario at 300 kW/m$^2$: (**a**) pressure at the outlet of channels 3; (**b**) pressure at the outlet of channels 4; (**c**) temperature at the outlet of channels 3; (**d**) temperature at the outlet of channels 4.

Considering all these sources of uncertainty it is clear that the reproduction of this phase is quite difficult to achieve even if the mathematical framework (BE meth. curves) suggests a quite good fit of the experimental trends. This discrepancy is probably due to the assumption of linear variations in the surrounding of the nominal values (see Equation (1)). Some parameters most likely have a cliff-edge effect, i.e., even small variations might lead to a completely different result. In these cases, an iterative process for the calibration is suggested [16], but this was not performed in the present study. Thus, the partial agreement found between the experimental and the computed results provides only a suggestion of the validity of the wall–coolant heat transfer, but not a guarantee.

The temperature evolution at the outlet of channels 2 and 9 during the same LOFA scenario is shown in Figures 22a and 22b, respectively. In these channels both the friction and the heat transferred from the heated area (conduction enclosure model) affected the

temperature evolution. To evaluate the influence of this last phenomenon the results from a run where the conduction enclosure model was switched off (BE run w/o Cond. curves) are included in the pictures. In this case, the coolant temperature was about 3–5 °C below the experimental figure, while when it was switched on a better agreement was found.

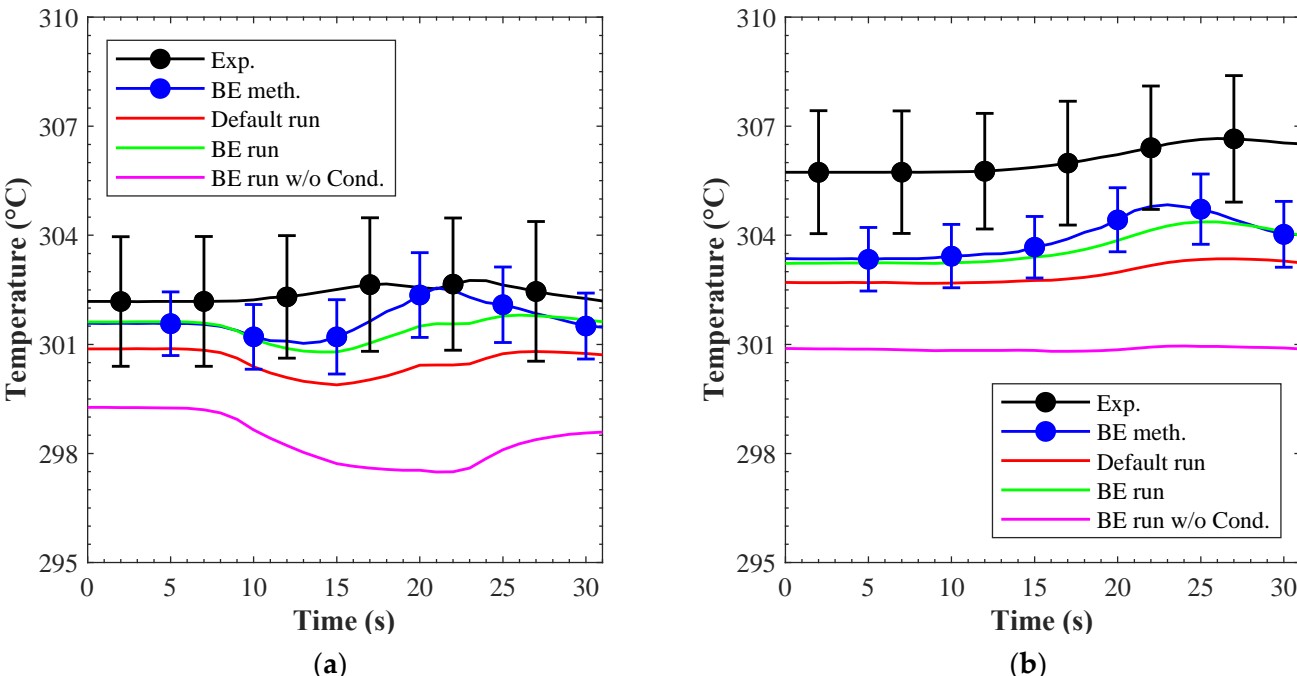

**Figure 22.** Coolant temperature for a 0% valve opening scenario at 300 kW/m² at the outlet of: (**a**) channel 2; (**b**) channel 9.

In turn, Figure 23 shows a comparison of the maximum temperature reported by the heat structures simulating the heated surface vs. the temperature limit of 550 °C. Six curves are reported: "Ch. 3" through "Ch. 8" refer to the BE run with the conduction enclosure model activated, while "Ch. 3 w/o Cond." and "Ch. 8 w/o Cond." refer to the BE run with the model switched off. When the model was on, the maximum temperature (channel 3) approached the safety limit, and a fraction of the incoming heat was transferred to the neighbor channels. Instead, when the model was switched off, the maximum temperature exceeded the temperature limit at about 40 °C, and no heat was transferred to the neighbor channels. Hence, in summary, the results shown in Figures 22 and 23 demonstrate: (a) the necessity of this model in these transients and (b) the validity of this model in the considered scenarios.

Finally, Figure 24 shows a bird's eye view comparison between the temperatures recorded by the IR camera (Figure 24a) and the temperature of the heat structures simulating the heated surface (Figure 24b). The black and red arrows in Figure 24b indicate the sides of the channel connected with the inlet and outlet nozzle, respectively. As stated in the experimental campaign section, the recordings of the IR camera were affected by quite a large uncertainty so they can only be used as a qualitative benchmark to assess the evolution of the surface temperature. The overall agreement found between the computed and the experimental results demonstrates again the capabilities of the conduction enclosure model and provide a further suggestion of the validity of the wall–fluid heat transfer.

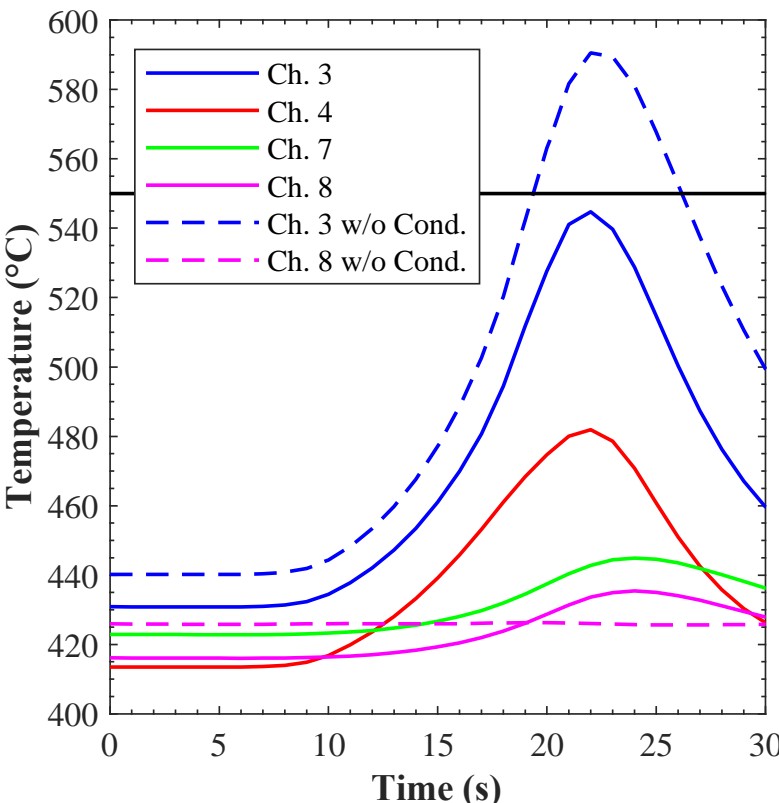

**Figure 23.** Maximum temperatures on the heat structures simulating the heated area—0% valve opening scenario at 300 kW/m$^2$.

The validity of the hydro-dynamic, the wall–fluid heat transfer, and the conduction enclosure models is also confirmed when looking at the results obtained for other LOFA scenarios. As an example, Figure 25 (for channels 3 and 4) and Figure 26 (for channels 2 and 9) show the results of the analysis conducted for a LOFA with 10% valve opening, while Figures 27 and 28 show similar results for the LOFA with 40% valve opening scenarios. For both cases the surface heat flux was 300 kW/m$^2$. In the 10% valve opening scenario, the temperature of the surface under heat load reached 550 °C, but no backflows were established through the channels under LOFA conditions, while in the 40% valve opening scenario the temperature stabilized below 550 °C. Under most aspects, the results of the 10% and 40% valve opening scenarios are similar to that of the 0% valve opening scenario:

- The outlet coolant pressure increased with time (Figure 25a,b) the velocity was reduced due to lower flow rates through the corresponding channels. No remarkable differences are shown between the default and the BE runs, and the "BE meth" curve shows a reduction in the standard deviation in concomitance with low coolant velocities.
- The outlet coolant temperatures from channels 3 and 4 (Figure 25c,d) showed a more scattered behavior than the outlet pressure. In channel 3 both the default and the BE runs followed the experimental trend, while in channel 4 the BE run the maximum temperature was not well captured, but the overall trend of the BE run was similar to the experimental one. This discrepancy can be linked to the high uncertainty affecting the heat load ($\pm$15%).
- The outlet coolant temperature from channels 2 and 9 (Figures 26 and 28) was well reproduced in both runs with the BE one closer to the experimental trend.

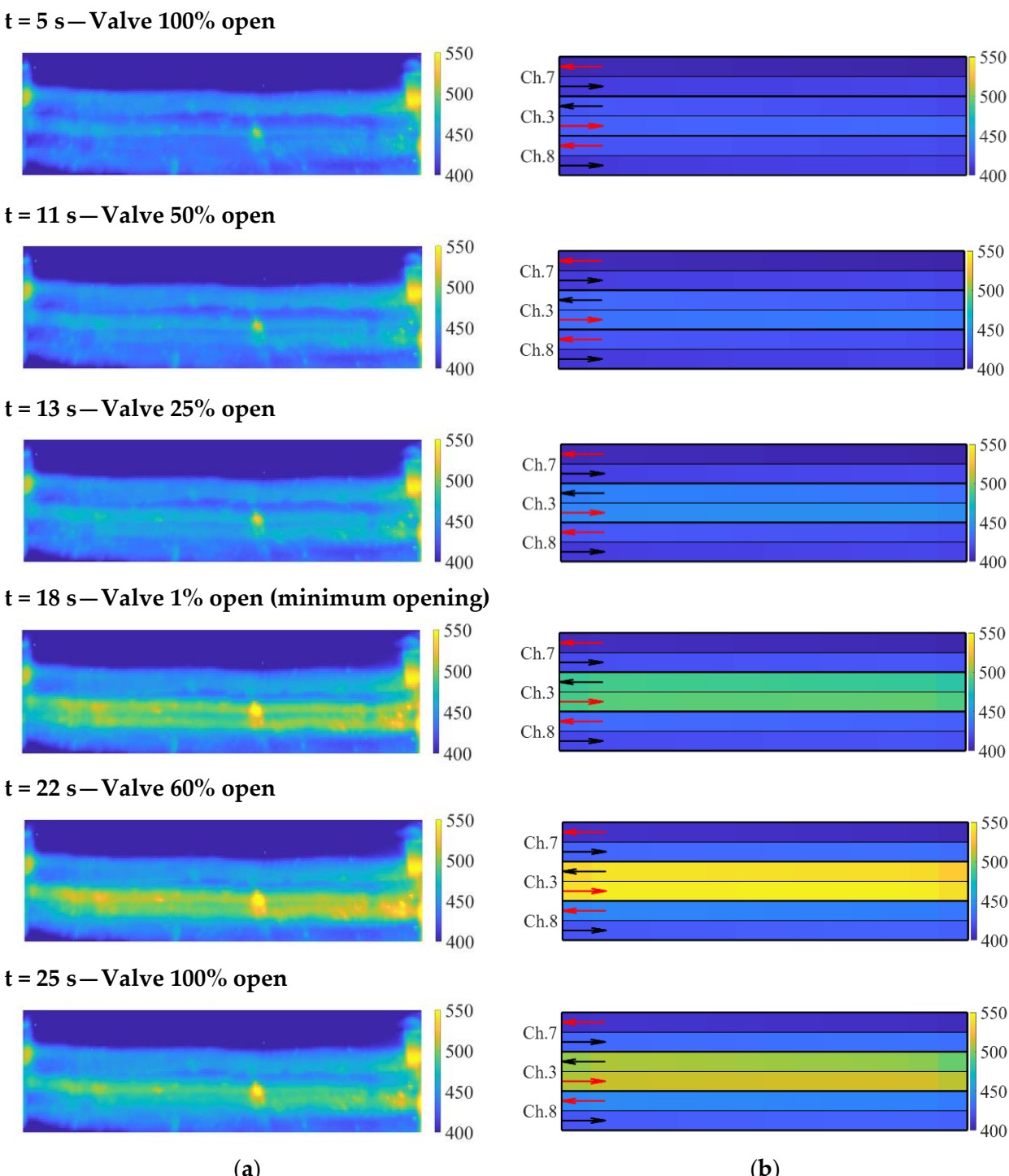

**Figure 24.** Temperature evolution of the mock-up surface under heat load for a 0% valve opening scenario at 300 kW/m$^2$: (**a**) IR camera-recorded temperatures during experiment at different times; (**b**) modeled and computed temperatures at the same time instants.

The good agreement (maximum discrepancy below 2%) shown for these last two scenarios give the final confirmation of the validity of the hydrodynamic model, the fluid-wall heat transfer, and the conduction enclosure model. With the exception of the backflows, the code shows a good agreement with the experimental trends, thus suggesting that the discrepancies in the 0% valve opening scenario were mostly related to user's assumptions and cliff-edge effects.

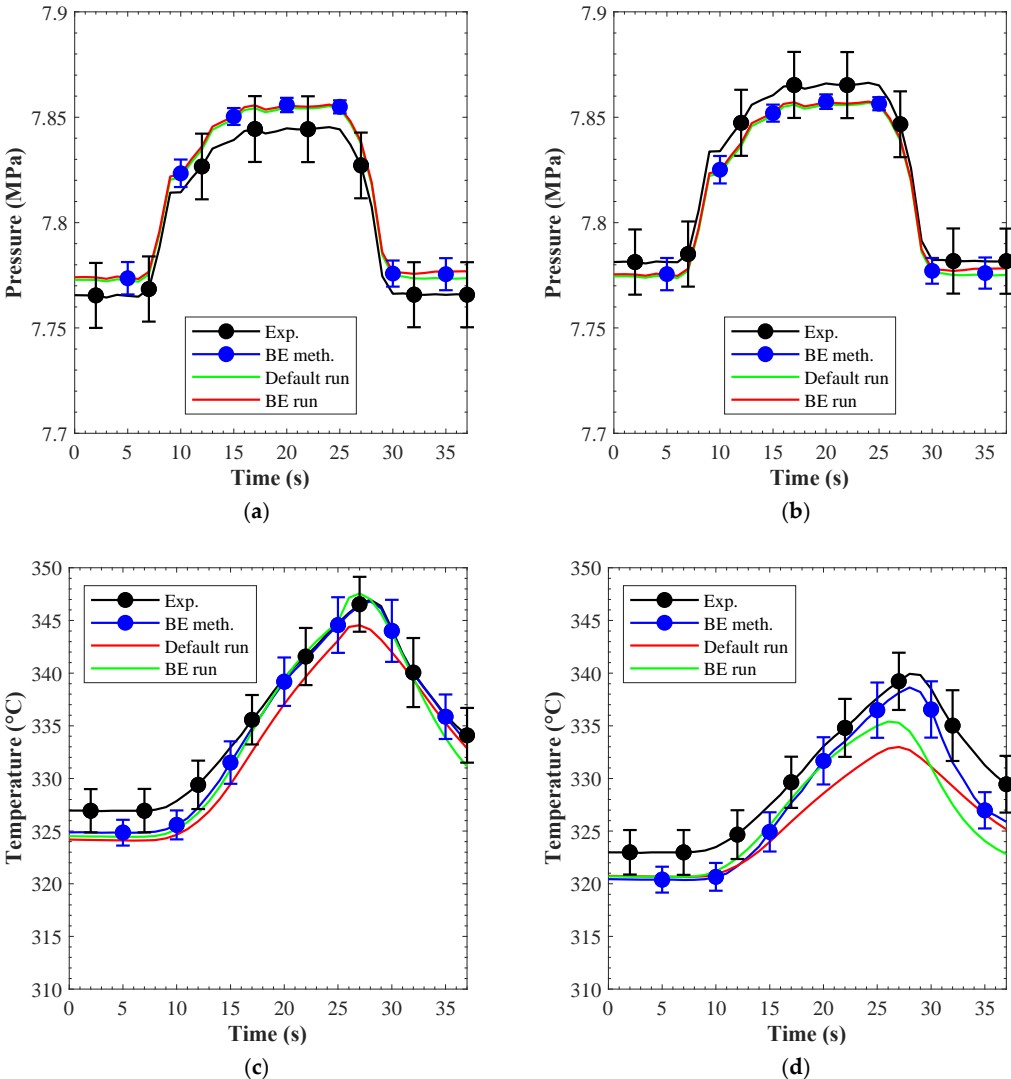

**Figure 25.** Helium parameters evolution during a 10% valve opening scenario at 300 kW/m$^2$: (**a**) pressure at the outlet of channels 3; (**b**) pressure at the outlet of channels 4; (**c**) temperature at the outlet of channels 3; (**d**) temperature at the outlet of channels 4.

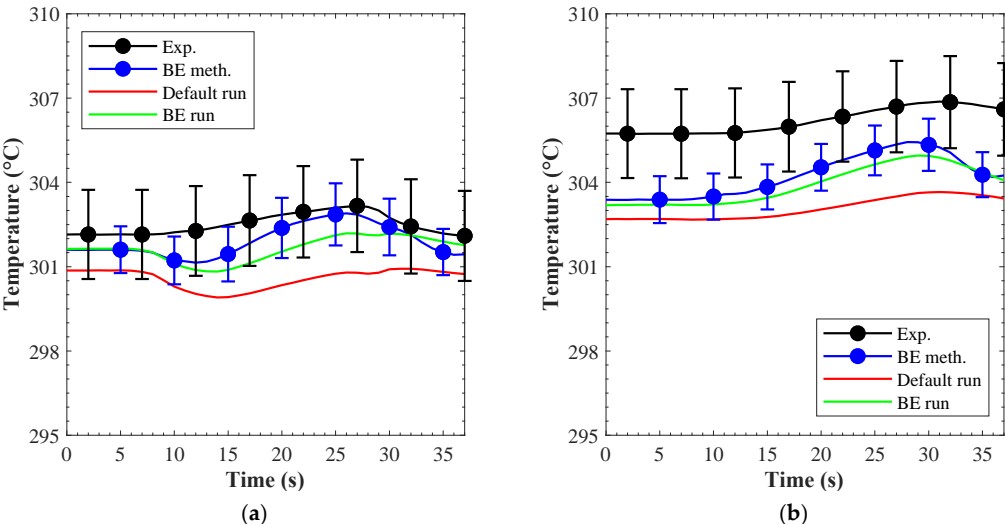

**Figure 26.** Outlet coolant temperature for a 10% valve opening scenario at 300 kW/m$^2$ at: (**a**) channel 2; (**b**) channel 9.

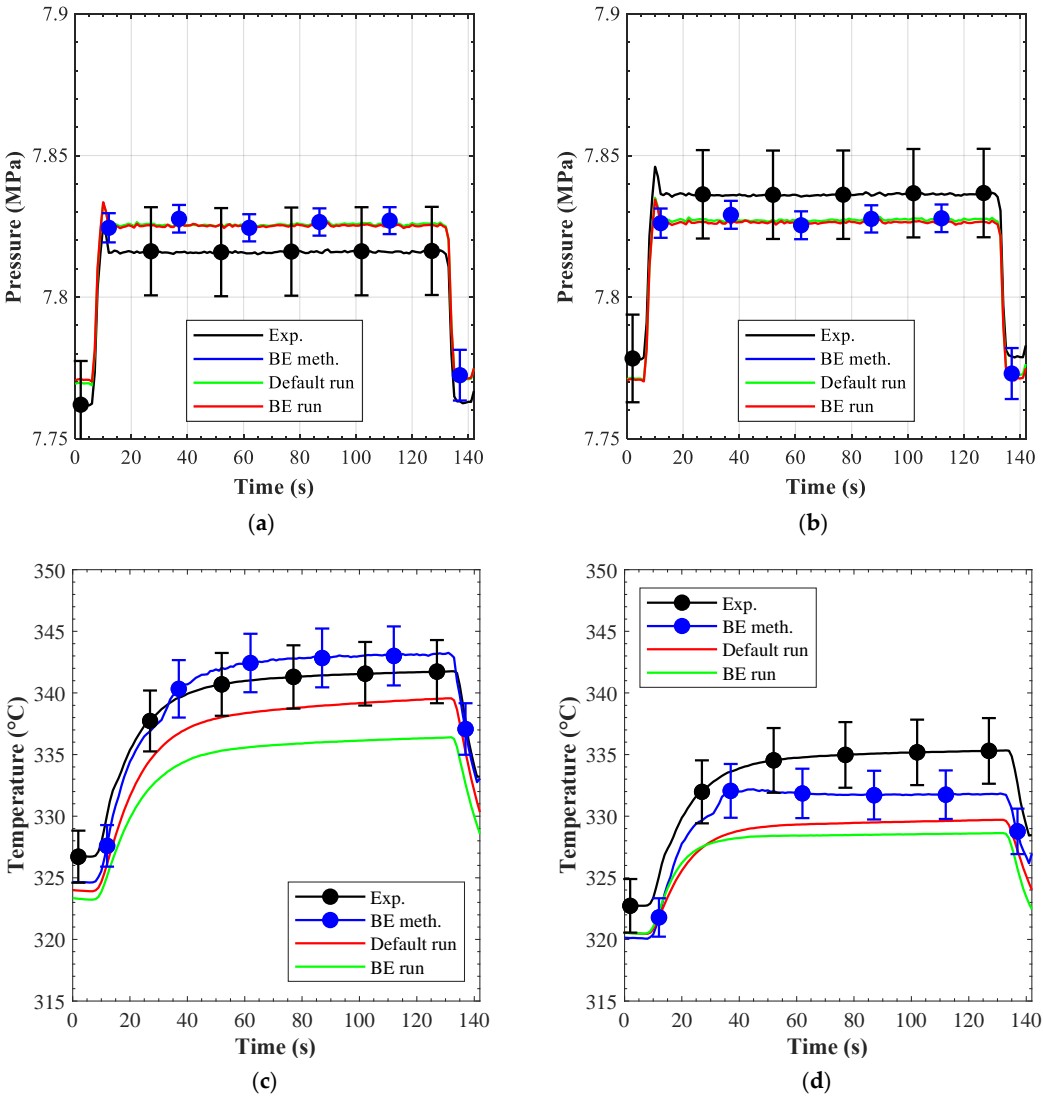

**Figure 27.** Helium parameters evolution during a 40% valve opening scenario at 300 kW/m$^2$: (**a**) pressure at the outlet of channels 3; (**b**) pressure at the outlet of channels 4; (**c**) temperature at the outlet of channels 3; (**d**) temperature at the outlet of channels 4.

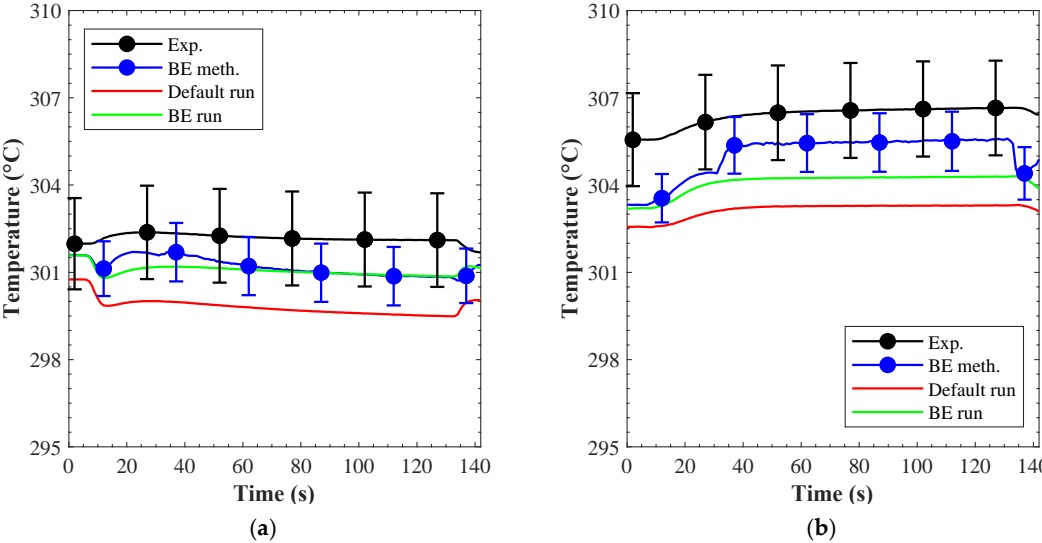

**Figure 28.** Outlet coolant temperature for a 40% valve opening scenario at 300 kW/m$^2$ in: (**a**) channel 2; (**b**) channel 9.

## 6. Conclusions

The present paper summarizes the findings of an experimental campaign investigating the behavior of a HCPB FW mock-up under different LOFA scenarios, including the pre- and post-test analysis that were performed. Thus, prior to the testing, several numerical studies were performed using both Ansys CFX for detailed CFD analysis and RELAP5-3D models for system-wide studies. The results of these analyses provided the basis for the definition of the tests matrix and the configuration of the experimental set-up.

The LOFA event in the mock-up was reproduced by reducing the flow rate in the respective branch when closing a valve on the outlet manifold of the channels 1–5. Six different scenarios were investigated:

- Valve 60% open;
- Valve 40% open;
- Valve 20% open;
- Valve 10% open;
- Valve 5% open;
- Valve fully closed (full LOFA).

Each scenario was performed for two different heat loads ($300\,\mathrm{kW/m^2}$ and $330\,\mathrm{kW/m^2}$) and each case was repeated several times to provide a robust set of data. The main outcomes of the experimental campaign were:

- The confirmation of the capability of the adopted coolant strategy to limit the heat excursion of the FW temperature if one of the branches suffers a partial LOFA;
- The demonstration that in the case of severe LOFA the FW temperature might exceed the safety limit in few seconds, thus hampering the integrity of the primary system. A prompt action of the plasma shutdown system would then be necessary to drive the system to a safe condition.

Finally, the data gathered from this experimental campaign were used for the validation of the RELAP5-3D code. A best-estimate methodology was also applied to ease the distinction between the user's assumptions and code limitations. Instead of simulating a specific case, "averaged" runs were investigated to identify the average behavior of the FWMU. The validation focused on three code physical models: the hydro-dynamic model, the fluid–wall heat transfer correlation (Gnielinski correlation [18] as implemented in RELAP5-3D [20]), and the conduction enclosure model. This activity reached the expected goals, proving the validity of these three models in DEMO-relevant conditions.

Under a more general point of view, the approach adopted to set-up, perform, and analyze the data of the experimental campaign was successful. The pre-test analysis gave fundamental insight into the set-up and the boundary conditions of the experimental campaign. In turn, the strategy adopted to execute the experimental tests allowed the creation of the robust "average" tests investigated in the validation activity. Then, this validation activity demonstrated the capabilities of the RELAP5-3D code in DEMO-relevant conditions of nuclear fusion application.

These activities also contributed to the creation of a relevant theoretical and practical experience that will be used in the on-going studies concerning incidental transients in real-plant scenarios.

**Author Contributions:** Conceptualization, B.-E.G., B.G., A.K., V.D.M.; methodology, M.I.-B., X.Z.J.; validation B.G.; writing—original draft preparation, B.-E.G.; writing—review and editing, B.G.; visualization, B.G.; supervision, M.I.-B., X.Z.J., R.S. All authors have read and agreed to the published version of the manuscript.

**Funding:** This work was carried out within the framework of the EUROfusion Consortium and received funding from the Euratom research and training programme 2014–2018 and 2019–2020 under grant agreement No 633053. The views and opinions expressed herein do not necessarily reflect those of the European Commission.

**Institutional Review Board Statement:** Not applicable.

**Informed Consent Statement:** Not applicable.

**Conflicts of Interest:** The authors declare no conflict of interest.

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
