# Peer review of "Experimental Investigation of a Helium-Cooled Breeding Blanket First Wall under LOFA Conditions and Pre-Test and Post-Test Numerical Analysis"

_applsci, doi:10.3390/app112412010_

Round 1

Reviewer 1 Report

Uncertainty analysis is missing: the authors only present the best estimate approach.

Grammar requires revision to avoid typos and the adoption somehow of more appropriate language

Author Response

Authors wish to thank the reviewer for comments received. Reviewer’s comments and revision have been taken into account and properly implemented in the manuscript.

Reviewer 2 Report

Paper Applsci-147715357

Title: Experimental investigation of a Helium-cooled Breeding Blan- 2 ket First Wall under LOFA conditions and pre-test and post-test 3 numerical analysis

I have some recommendations/questions for the authors, before to accept the manuscript as Applied Science’s paper.

(i) In Abstract, the authors have stated the following: “All these activities contributed to the creation of a relevant theoretical and practical experience that can be further used in the set-up and execution of the future experimental campaigns”.

It is necessary to complete the Abstract presenting a short commentary concerning the actual state of art where the present research will be inserted.

(ii) The authors have only revised the following paper published in Applied Sciences:

“16. Gonfiotti, B.; Angelucci, M.; Ghidersa, B.-E.; Jin, X. Z.; Ionescu-Bujor, M.; Paci, S.; Stieglitz, R. Best-Estimate for System Codes 857 (BeSYC): a new soft-ware to perform Best-Estimate Plus Uncertainty analyses with thermal-hydraulic and safety system codes. 858 Appl. Sci. 2021, 11, x. https://doi.org/10.3390/xxxxx”.

Why? Are there other publications about the research topic in Applied Sciences? That key point needs to be clarified by authors to justify the publication. Furthermore, the section 1 depends on concise and suitable discussion about previous works published in Applied Sciences.

(iii) In section 3, a Table must be included to summarize the main quantities experimentally measured and their uncertainties.

(iv) The sections referring to numerical approach must be organized as section 4, i.e. they must be separated of section 3 about experimental approach.

(v) What is the convergence criterion (mesh refinement details) for the numerical results presented in Figure 3?

(vi) Figure 5 needs to be better explained and interpreted into the manuscript. In my opinion, there are lots of results to be understood and this has turned the reading a few tiring. In general, figures and tables should be better interpreted including physical sense in some test cases. I would like to recommend a suitable strategy to discuss the more relevant results. What are the more representative results and contributions? That is my main concern with the present manuscript. I hope that the authors can solve that situation in due time.

(vii) In section 8, a perspective for future research is welcome.

Author Response

Authors wish to thank the reviewer for comments received. Reviewer’s comments have been taken into consideration and implemented in the manuscript.

Round 2

Reviewer 2 Report

Paper Applsci-1477153-R

Title: Experimental investigation of a Helium-cooled Breeding Blanket First Wall under LOFA conditions and pre-test and post-test numerical analysis

The original text of the manuscript has been satisfactory revised. In my opinion, the manuscript can be published as an Applied Science’s paper.